# Investigations on Modularity and Invariance for Compositional Robustness

## Abstract

By default neural networks are not robust to changes in data distribution. This has been demonstrated with simple image corruptions, such as blurring or adding noise, degrading image classification performance. Many methods have been proposed to mitigate these issues but for the most part models are evaluated on single corruptions. In reality, visual space is compositional in nature, that is, that as well as robustness to elemental corruptions, robustness to compositions of corruptions is also needed. In this work we develop a compositional image classification task where, given a few elemental corruptions, models are asked to generalize to compositions of these corruptions. That is, to achieve *compositional robustness*. We experimentally compare empirical risk minimization with an invariance building pairwise contrastive loss and, counter to common intuitions in domain generalization, achieve only marginal improvements in compositional robustness by encouraging invariance. To move beyond invariance, following previously proposed inductive biases that model architectures should reflect data structure, we introduce a modular architecture whose structure replicates the compositional nature of the task. We then show that this modular approach consistently achieves better compositional robustness than non-modular approaches. We additionally find empirical evidence that the degree of invariance between representations of 'in-distribution' elemental corruptions fails to correlate with robustness to 'out-of-distribution' compositions of corruptions.

## 1 Introduction

Biologically intelligent systems show a remarkable ability to generalize beyond their training stimuli, that is to learn new concepts from no, or few, examples by combining previously learned concepts (Ito et al., 2022; Lake et al., 2019; Piantadosi & Aslin, 2016; Schulz et al., 2016). In contrast, artificial neural networks are surprisingly brittle, failing to recognize known categories when presented with images with fairly minor corruptions (Dodge & Karam, 2017; Geirhos et al., 2021; Hosseini et al., 2017; Hendrycks & Dieterich, 2019; Jang et al., 2021). Many methods have been proposed for learning more robust representations, including data augmented training techniques (Hendrycks et al., 2020; 2021; Jang et al., 2021; Yun et al., 2019; Zhang et al., 2018), and encouraging invariant representations or predictions (Huang et al., 2022; Kim et al., 2019; Sinha & Dieng, 2021; Von Kügelgen et al., 2021). However, when the robustness of these methods is evaluated it tends to be on single corruptions of the type seen in ImageNet-C (Hendrycks & Dieterich, 2019).

In reality, the space of possible corruptions is compositional. If we draw a loose correspondence between corruptions and real world weather conditions, with noise akin to rain on a windshield, blur as fog and a contrast change as a change in brightness, we see it is in fact possible to have rain, fog and bright sun simultaneously. In this work we extend the notion of robustness over corruptions to robustness over *compositions of corruptions*. We construct a compositional image classification task where a neural network is trained on single *elemental corruptions* and evaluated on *compositions* of these corruptions (Figure 1). Importantly, this is not an adversarial or no-free-lunch task, as we want the AI systems we develop to be capable of compositional generalization (Bahdanau et al., 2019b; Chalmers, 1990; Fodor & Pylyshyn, 1988; Goyal & Bengio, 2022; Lake & Baroni, 2018; Lake et al., 2017; Mendez & Eaton, 2022).

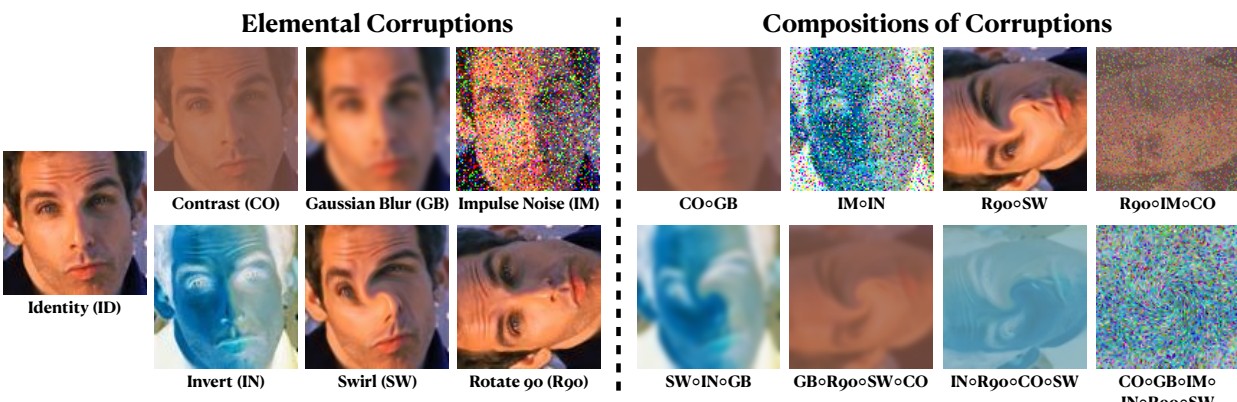

Figure 1: The compositional robustness task. A model is trained jointly on images corrupted with elemental corruptions (left) and evaluated on images corrupted with compositions of these corruptions (right). Shown is *all* 7 elemental corruptions and a *subset* of the 160 compositions of corruptions.

If natural visual data can be decomposed into a set of elemental functions (or mechanisms (Peters et al., 2017; Parascandolo et al., 2018)), we do not yet know how to find them. The compositional robustness task we create allows us to experiment with a compositional structure where the underlying elemental functions are known. By studying the behaviors of neural networks under this structure, we aim to gain insights into how we might develop methods for better compositional robustness. Such insights could be applied to create systems that generalize more robustly or allow for lower data collection costs, needing only to collect or synthesize the elemental corruptions instead of the exponentially large number of compositions. Finally, this task creates a new domain generalization task on which we can evaluate the generality of proposed methods for domain generalization. In domain generalization parlance, a system is trained on data from multiple training domains (the elemental corruptions), and then evaluated on data from a related set of test domains (the compositions), from which no data samples are seen during training.

To better understand how neural networks behave on out-of-distribution compositional data we evaluate different methods for domain generalization on this task. Firstly, we explore empirical risk minimization (ERM), which has been shown to be a strong baseline when correctly tuned (Gulrajani & Lopez-Paz, 2022). Secondly, we evaluate a setup where invariance between the same image under different corruptions is explicitly encouraged using the contrastive loss (Chen et al., 2020; Gutmann & Hyvärinen, 2010; Hadsell et al., 2006), since a central theme in domain generalization has been to encourage the learning of invariant representations (Ahmed et al., 2021; Albuquerque et al., 2019; Arjovsky et al., 2019; Dou et al., 2019; Li et al., 2018b; Ghifary et al., 2015; Kim et al., 2021; Li et al., 2018a; Motiian et al., 2017; Sakai et al., 2022; Creager et al., 2021). Finally, we introduce a modular architecture to better reflect the compositional structure of the task (Pfeiffer et al., 2023). Here, rather than all parameters jointly modelling all corruptions, each elemental image corruption is 'undone' by a separate module in latent space.

Counter to our initial expectations we find that training to encourage invariant representations with the contrastive loss offers only minor improvements in terms of out-of-distribution accuracy, whilst the modular architecture consistently outperforms other methods. Additionally, we find that the degree of invariance between representations of elemental corruptions fails to correlate with performance on out-of-distribution compositions of corruptions. At their narrowest interpretation, these results empirically show that for compositional robustness, when training domains consist only of the elemental components, modular approaches tend to outperform monolithic (non-modular) approaches. At their broadest interpretation our results question whether encouraging non-trivially[1] invariant representations is sufficient to achieve compositional domain generalization. This indicates that there is still work to be done on understanding the additional properties required for compositional robustness and suggests more modular architectures as a promising candidate for one such property.

---

[1]The trivial case, with constant representations, has maximal invariance but cannot achieve good generalization.

## 2 Related Work

We now briefly recap related works from the areas of domain generalization, invariant representations, modularity, compositional generalization and robustness.

**Domain Generalization and Invariant Representations.** The creation of models that are robust to unseen changes in data distribution is the work of domain generalization. Given certain training domains, the aim of domain generalization is to build models that can generalize to related unseen test domains. One common approach is to encourage the learning of invariant representations between training domains whilst achieving high performance (Ahmed et al., 2021; Albuquerque et al., 2019; Arjovsky et al., 2019; Dou et al., 2019; Ghifary et al., 2015; Kim et al., 2021; Li et al., 2018a;b; Motiian et al., 2017; Sakai et al., 2022; Creager et al., 2021), with the idea that this will lead to invariant representations between training and test domains and hence good generalization performance. However, this relies on an implicit assumption that we have sufficient training domains that are reasonably representative samples from some meta-distribution of domains (this has been made explicit in some works Krueger et al. (2021); Eastwood et al. (2022c)). It is not clear that this will be true in general, and arguably replaces the problematic assumption of $i.i.d$ data with an equally problematic assumption of $i.i.d$ domains. What's more, such generalist approaches may be unable to take structure amongst training domains into account. It should be noted that there has also been substantial work on encouraging invariance for the related task of domain adaptation where (unlabelled) data from test domains is available (Ganin et al., 2016; Tzeng et al., 2017; Sun & Saenko, 2016; Long et al., 2015; 2018; Eastwood et al., 2022b). Despite being motivated by theoretical work (Ben-David et al., 2007; 2010), the central role of invariance in domain adaptation and generalization has been questioned (Zhao et al., 2019; Johansson et al., 2019; Rosenfeld et al., 2021; Shen et al., 2022; Akuzawa et al., 2020). In Section 4.3 we discuss the limitations of encouraging invariance for compositional robustness.

**Relational Inductive Biases and Modularity.** A closely related approach to learning robust representations aims to take advantage of explicit structure in data. These relational inductive biases (Battaglia et al., 2018) aim to include knowledge about entities and the relations between them into neural network architectures. For example, we can encode that entities should not change under certain transformations by building invariance to these transformations into our architectures. Work on equivariance beyond translation explicitly creates such robustness (Cohen & Welling, 2016; Weiler et al., 2018; Worrall et al., 2017) but is usually formulated in terms of group actions (Cohen et al., 2019) so is limited to invertible transformations. More general approaches aim to uncover structure by decomposing data into independent (causal) mechanisms (Peters et al., 2017; Parascandolo et al., 2018; Schölkopf et al., 2021; Goyal et al., 2021; Goyal & Bengio, 2022) or disentangled factors of variation (Chen et al., 2016; Higgins et al., 2017; Kim & Mnih, 2018; Roth et al., 2023; Eastwood & Williams, 2018; Locatello et al., 2019; Schott et al., 2022; Montero et al., 2021; 2022). Ways to explicitly model decomposable structures in data include pre-training on primitive components (Ito et al., 2022) and using modular architectures to encode structure (Jaderberg et al., 2015; Andreas et al., 2016a;b; D'Amario et al., 2021; Bahdanau et al., 2019b; Goyal et al., 2021; 2022; Mendez & Eaton, 2021; Madan et al., 2022; Carvalho et al., 2023; Pfeiffer et al., 2023). In contrast, in this work we know how the data structure decomposes and explore the performance of modular and non-modular architectures on the recomposition of known elemental components.

**Compositional Generalization.** The visual world is compositional (Bahdanau et al., 2019a; Lake et al., 2015; 2017; Romaszko et al., 2017; Krishna et al., 2017). Whilst much has been made of compositionality in language (linguistic compositionality) and reasoning (conceptual compositionality) (Andreas et al., 2016a; Battaglia et al., 2018; Furrer et al., 2020; Hu et al., 2017; Johnson et al., 2017; Lake & Baroni, 2018; Lake et al., 2017; Liška et al., 2018; Mendez & Eaton, 2022; Qiu et al., 2022; Xie et al., 2022; Schmidhuber, 1990), compositional robustness has received relatively little attention. Recent AI systems still fail on compositional tasks (Keysers et al., 2020; Lake & Baroni, 2018; Schott et al., 2022; van der Velde et al., 2004) where the space of generalization grows exponentially with the number of elemental components. Whilst practically it is not possible to sample all combinations of elemental components, one interpretation of large models (Geirhos et al., 2021; Kaplan et al., 2020; Radford et al., 2021) is that they aim to sample densely enough to generalize to unseen combinations. However, for real world data, it is unclear how big the compositional space is and how densely we need to sample, with this being particularly pertinent if the data distribution is high-dimensional (Geiger et al., 2020; Wainwright, 2019) or fat tailed (Taleb, 2020). To that

end, several works have analyzed controlled settings, aiming to understand the best settings for training in order to achieve the best generalization (Ahmed et al., 2021; Cooper et al., 2021; Schott et al., 2022).

**Robustness Over Image Corruptions.** Whilst the aforementioned work aims to improve the robustness of neural networks, many have worked specifically on improving robustness for common image corruptions and adversarial examples (Hendrycks & Dietterich, 2019; Jang et al., 2021; Hendrycks et al., 2020; 2021; Yun et al., 2019; Zhang et al., 2018; Huang et al., 2022; Kim et al., 2019; Sinha & Dieng, 2021; Von Kügelgen et al., 2021; Baidya et al., 2021). However, the majority of previous works are evaluated only on single corruptions, ignoring the true compositional space formed by the corruptions.

## 3    Methods

We now describe the methodology we use to investigate robustness to compositions of image corruptions. We begin by creating a framework for evaluation and then describe the different approaches we investigate including training with ERM, encouraging invariance and building modular architectures.

### 3.1    A Framework for Evaluating Compositional Robustness

We design a framework for evaluating compositional robustness on any dataset for image classification. We first create elemental components by applying six different corruptions separately to all images. These corruptions along with the original, *Identity (ID)*, data create 7 training domains. We use the corruptions *Contrast (CO)*, *Gaussian Blur (GB)*, *Impulse Noise (IM)*, *Invert (IN)*, *Rotate 90° (R90)* and *Swirl (SW)*, seen in Figure 1 (left). We choose these corruptions to include a mixture of long-range and local effects as well as invertible and non-invertible corruptions. A further exploration of the choice and parameter settings of corruptions is given in Appendix A.

To test compositional robustness we create images from compositions of the elemental corruptions, see Figure 1 (right). We consider every possible permutation of compositions of two corruptions (excluding *Identity*) giving $^6P_2 = 30$ possible compositions. For compositions of more than two corruptions we sample the possible permutations to approximately balance the contributions of compositions containing different numbers of elemental corruptions (the sampling process is described in Appendix A). This creates 40 possible compositions of 3 corruptions and 30 possible compositions for each of 4, 5 and 6 corruptions. Altogether the compositions form 160 test domains. The task we then try to solve is to achieve the highest classification accuracy on images from the 160 compositional test domains whilst training only on the 7 elemental training domains.

### 3.2    Monolithic Approaches

A domain generalization task consists of data from related domains or environments $\mathcal{D}_e = \{(\boldsymbol{x}_e^{(i)}, y_e^{(i)})\}_{i=1}^{N_e}$, with $e \in \mathcal{E}_{all}$, where $\mathcal{E}_{all}$ is the set of all domains we wish to generalize to and $N_e$ the number of datapoints in domain $e$. However, during training we only have access to a subset of domains $\mathcal{E}_{tr} \subset \mathcal{E}_{all}$. For our task, $\mathcal{E}_{tr}$ is the set of elemental training domains, $|\mathcal{E}_{tr}| = 7$, and $\mathcal{E}_{all}$ additionally includes the compositional test domains, $|\mathcal{E}_{all}| = 167$. As we use the same set of base images to create corrupted images, the number of datapoints, $N_e$, is the same across all domains.

For a neural network $f_{\boldsymbol{\theta}}$ parameterized by $\boldsymbol{\theta}$, we aim to find parameters, $\boldsymbol{\theta}^*$, from parameter space $\boldsymbol{\Theta}$, that optimize loss function $\mathcal{L}$, on training domains $\mathcal{E}_{tr}$. The accuracy of $f_{\boldsymbol{\theta}*}$ is then evaluated on the test domains. Monolithic approaches share all parameters, $\boldsymbol{\theta}^*$, over all domains where,

$$\boldsymbol{\theta}^* = \operatorname*{argmin}_{\boldsymbol{\theta} \in \boldsymbol{\Theta}} \sum_{e \in \mathcal{E}_{tr}} \sum_{i=1}^{N_e} \mathcal{L}(f_{\boldsymbol{\theta}}(\boldsymbol{x}_e^{(i)}, y_e^{(i)})). \tag{1}$$

The first approach we evaluate is Empirical Risk Minimization (ERM), training all parameters jointly to minimize some risk function over training domains. We set $\mathcal{L}$ to be the mean cross entropy loss.

The second approach we evaluate is contrastive training. A standard domain generalization approach is to encourage invariance between representations on the training domains (Zhou et al., 2022) and since we

have paired data between domains we can explicitly encourage invariance using the contrastive loss (Chen et al., 2020; Gutmann & Hyvärinen, 2010; Hadsell et al., 2006). Note that the availability of paired data creates a best-case set up for the learning of invariant representations and that learning a representation that is invariant for paired images from different domains would satisfy the invariance encouraging criteria of previous works (Arjovsky et al., 2019; Li et al., 2018b; Dou et al., 2019).

We follow the SimCLR contrastive training formulation (Chen et al., 2020), taking $B$ datapoints from each elemental training domain (created from the same base images) to get a minibatch of size $B|\mathcal{E}_{tr}|$. Applying an additional index to each of the domains in $\mathcal{E}_{tr}$ to get $\mathcal{E}_{tr} = \{e_d\}_{d=1}^D$, positive pairs come from pairs of the same image under different corruptions $(\boldsymbol{x}_{e_r}^{(i)}, \boldsymbol{x}_{e_s}^{(i)}), r \neq s$, and negative pairs from all other pairs in the minibatch $(\boldsymbol{x}_{e_r}^{(i)}, \boldsymbol{x}_{e_s}^{(j)}), i \neq j$. We apply the contrastive loss on representations from the penultimate layer of $f_{\boldsymbol{\theta}}$, notating the representation for $\boldsymbol{x}_e^{(i)}$ as $\boldsymbol{z}_e^{(i)}$. Using cosine similarity, $\text{sim}(\boldsymbol{u}, \boldsymbol{v}) = \boldsymbol{u}^T \boldsymbol{v} / \|\boldsymbol{u}\| \|\boldsymbol{v}\|$, to measure similarity between representations we define the loss for a positive pair in the minibatch as

$$\ell(\boldsymbol{x}_{e_r}^{(i)}, \boldsymbol{x}_{e_s}^{(i)}) = -\log \frac{\exp(\text{sim}(\boldsymbol{z}_{e_r}^{(i)}, \boldsymbol{z}_{e_s}^{(i)})/\tau)}{\sum_{d=1}^D \sum_{k=1}^B \mathbb{1}[k \neq i] \exp(\text{sim}(\boldsymbol{z}_{e_r}^{(i)}, \boldsymbol{z}_{e_d}^{(k)})/\tau)}, \tag{2}$$

where $\tau$ is a temperature parameter and $\mathbb{1}[k \neq i]$ is an indicator function equal to 1 when $k \neq i$ and 0 otherwise. We compute this loss across all positive pairs in the minibatch to encourage invariant representations. To learn to classify, we additionally include the cross entropy loss to arrive at,

$$\mathcal{L}(f_{\boldsymbol{\theta}}(\boldsymbol{x}_{e_r}^{(i)}, y_{e_r}^{(i)})) = -\sum_{c=1}^C \mathbb{1}[y_{e_r}^{(i)} = c] \log(\sigma(f_{\boldsymbol{\theta}}(\boldsymbol{x}_{e_r}^{(i)}))_c + \lambda \sum_{\substack{e_s \in \mathcal{E}_{tr} \\ s \neq r}} \ell(\boldsymbol{x}_{e_r}^{(i)}, \boldsymbol{x}_{e_s}^{(i)}). \tag{3}$$

Here the first term is the cross entropy loss, with $C$ the total number of categories, $\mathbb{1}[y = c]$ an indicator function that is 1 when $y = c$ and 0 otherwise, $\sigma$ the softmax operation, and, in a slight overloading of notation, subscript $c$ represents the $c^{th}$ entry of the log-softmax vector. $\lambda$ is a hyper-parameter weighting the influence of the cross entropy and contrastive terms. Note also, as described above, Equation 3 is calculated on a minibatch rather than over all datapoints simultaneously.

To evaluate the monolithic approaches on compositions of corruptions we simply calculate classification accuracy on the domains in $\mathcal{E}_{all}$.

## 3.3 A Modular Approach

The final approach we evaluate is a modular architecture, as it has been argued that modularity is a key feature of robust, intelligent systems (Goyal & Bengio, 2022; Mahowald et al., 2023). For each elemental corruption we add one module to our network which aims to 'undo' the corruption in latent space. In practice these modules are intermediate layers that operate on hidden representations to map the representation of a corrupted image to the representation of the same image when uncorrupted. To make this possible modules are designed to have input and output features with the same shape. When classifying a test image corrupted with a composition of elemental corruptions we sequentially apply the modules for each corruption present in the composition. For example, if we are testing on the composition *IN∘GB* we apply both the module trained on the *Invert* corruption and the module trained on the *Gaussian Blur* corruption. Modules that are located in-between earlier layers of the network are applied first, if modules are in the same layer we apply the module which appears first in the permutation ordering (Section 3.1).

To formalize this idea, we split network parameters $\boldsymbol{\theta}$ into one set of parameters shared over all domains, $\boldsymbol{\theta}_{shared}$, and an additional set of domain specific module parameters for each training domain $\{\boldsymbol{\theta}_e\}_{e \in \mathcal{E}_{tr}}$, similar to residual adaptation (Rebuffi et al., 2017; 2018). In practice $\boldsymbol{\theta}_{shared}$ parameterizes a neural network and $\boldsymbol{\theta}_e$ the intermediate layers that can be inserted when working with domain $e$.

To train this system we first train parameters $\boldsymbol{\theta}_{shared}$ on *Identity* data using the cross entropy loss. We then freeze $\boldsymbol{\theta}_{shared}$ and train separate modules parameterized by $\boldsymbol{\theta}_e$ on data from each elemental training domain $e \in \mathcal{E}_{tr}$ along with paired *Identity* data. Since we encourage the modules to 'undo' corruptions, we use the loss function from Equation 3 with minor modifications. Firstly, the set of domains for the contrastive loss is

limited to only the relevant elemental training domain and the *Identity* domain. Secondly, for the *Identity* data, latent representation $z$ is from the layer at which the module is inserted and for the corrupted data from the output of the module, spatially flattening the feature map if required (as opposed to from the penultimate network layer as described when introducing Equation 2). Appendix B contains a graphical depiction of this process.

An important design choice for any modular approach is how to choose where to locate the modules, with recent works observing that different domain changes should be dealt with in different neural network layers (Eastwood et al., 2022a; Royer & Lampert, 2020; Lee et al., 2023). We take a very simple automatic approach, training separate modules between each layer of the network parameterized by $\boldsymbol{\theta}_{shared}$ for 5 epochs. We then select the module with the best in-distribution accuracy on a held-out validation set as the module to train to completion. This is similar to using adaptation speed (Bengio et al., 2019; Le Priol et al., 2021) as a proxy to discover modular decompositions, although in practice we find if we use adaptation times substantially smaller than 5 epochs we can erroneously select module locations that do not achieve optimal in-distribution performance.

### 3.4 Measuring the Invariance of Learned Representations

Since encouraging invariance is a prominent theme in the domain generalization literature (Zhou et al., 2022) we also empirically investigate the role of invariant representations in generalizing to unseen compositions of corruptions. We create two invariance scores following the methods of Madan et al. (2022).

For every neuron in the penultimate layer of a network (after applying modules if applicable) we calculate the mean activation per domain-category pair over all test data. The activations are normalized by the maximum firing of the neuron over all domains with any dead neurons (with maximum firing less than $10^{-6}$) discarded. For a specific test domain (a specific composition of corruptions) we select only the domain-category pairs where the domain is either the test domain itself or one of the elemental corruptions used to create the composition for the test domain. For each neuron, this creates an activation grid with a column-count equal to the number of categories in the dataset and row-count equal to the number of elemental corruptions in the composition plus one row for the composition itself. An example grid is shown in Appendix C, Table 3.

This activation grid is then normalized again so that all values lie between 0 and 1 by subtracting the minimum value in the grid from every cell and dividing by the difference between the maximum and minimum values. We notate the activation values by $a_{i,j}$, with $i$ referencing the domain and $j$ the category. Additionally we take the number of elemental corruption domains in the grid to be indexed $1, \ldots, E$ and the composition to have index $E + 1$, that is, $i \in \{1, \ldots, E + 1\}$. Taking the view that neurons can be interpreted as feature detectors (Olah et al., 2020; Sarkar et al., 2023), we select the preferred category, $j^*$, on the training domains as the category for which the neuron maximally activates, $j^* = \mathrm{argmax}_j \sum_{i=1}^{E} a_{i,j}$. We then calculate the *elemental invariance score*, $I_e$ as the maximum difference in activations amongst the elemental corruptions, with the idea that this score should be high when all elemental corruptions activate the neuron in a similar way. We additionally calculate the *composition invariance score*, $I_c$, which measures how similarly the neuron activates on the composition compared to the closest elemental corruption.

$$I_e = 1 - (\max_i a_{i,j^*} - \min_i a_{i,j^*}), \qquad I_c = 1 - \min\{|a_{i,j^*} - a_{E+1,j*}|\}_{i=1}^{E}. \qquad (4)$$

These scores always lie between 0 and 1, with higher numbers representing more invariant representations. We calculate these scores for every neuron in the penultimate layer of the network and report the median scores over all (non-dead) neurons in our results. We refer the reader to Madan et al. for further motivation and details or Appendix C for an illustrative example.

### 3.5 Datasets, Architectures and Training Procedure

We evaluate each training approach on three different datasets for image classification: EMNIST (Cohen et al., 2017), an extended MNIST with 47 handwritten character classes; CIFAR-10 (Krizhevsky et al., 2009), a simple object recognition dataset with 10 classes, and FACESCRUB (Ng & Winkler, 2014), a face-recognition

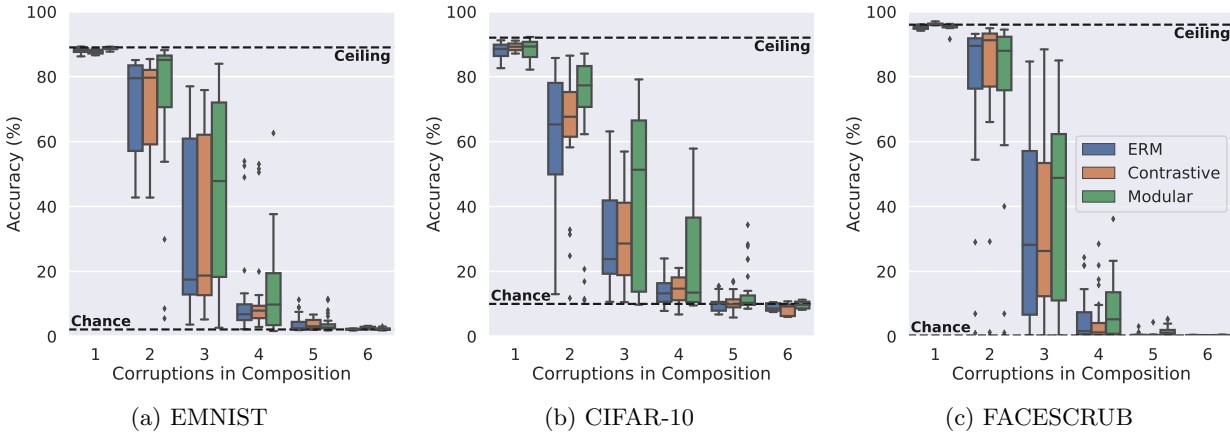

Figure 2: Evaluating compositional robustness on different datasets. Evaluation domains are divided into groups depending on the number of elemental corruptions making up a composition. Different colored boxes (left to right in each triple) show the performance of ERM, contrastive training, and the modular approach. Ceiling accuracy is determined by a model trained and tested on *Identity* data.

dataset. For FACESCRUB we follow (Vogelsang et al., 2018) removing classes with fewer than 100 images, resulting in 388 classes, with each class representing an individual identity. We train using stochastic gradient descent with momentum 0.9 and weight decay $5 \times 10^{-4}$, learning rate is set using a grid search over $\{1, 10^{-1}, 10^{-2}, 10^{-3}\}$ and contrastive loss weighting, $\lambda$, over $\{10, 1, 10^{-1}, 10^{-2}\}$, with the best setting selected based on the performance on a validation set of the *training* domains (Gulrajani & Lopez-Paz, 2022). $\tau$ from Equation 2 is set to 0.15 in all experiments. We use a batch size of 256 (or the nearest multiple of $|\mathcal{E}_{tr}|$ for the contrastive loss) and train for a maximum of 200 epochs, using early stopping on the held out validation set. Each experiment is run over three randomly seeded initializations. CIFAR-10 and FACESCRUB images are augmented with random cropping and flipping, ensuring positively paired examples receive exactly the same augmentation. For EMNIST we use a simple convolutional network with a LeNet-like (LeCun et al., 1998) architecture with modules made up from convolutional layers. For CIFAR-10 we use ResNet18 (He et al., 2016) without the first max pooling layer, wherever possible using ResNet blocks as modules. For FACESCRUB we use Inception-v3 (Szegedy et al., 2016) without the auxiliary classifier. As with ResNet we use additional Inception-v3 layers as modules wherever possible. For full architectural details see Appendix D.

## 4 Results

In this section we evaluate the compositional robustness of the different training approaches, first by examining the accuracy of different methods on unseen compositions of corruptions. We additionally explore the relationship between compositional robustness and invariance amongst representations of elemental corruptions. We end on the practical limitations of the approaches we consider in this study.

### 4.1 Monolithic Approaches Show Limited Compositional Robustness

Figure 2 shows the classification accuracies of each of the three approaches for each of the three datasets. The evaluation domains, $\mathcal{E}_{all}$, are divided into groups depending on how many elemental corruptions are in the composition applied to images in a domain. Across all methods and datasets we see domains with 1 corruption achieve very good, near ceiling, performance. This is not surprising as this represents the accuracy on the elemental training domains. A granular view for each of the 167 domains for every method can be seen in heat maps in Appendix G.

In Figure 2 the blue and orange box plots show the performance of ERM and contrastive training respectively, for which we can observe some general trends. Firstly, accuracy on compositions drops as the number of elemental corruptions in a composition increases, with compositions of 5 or 6 corruptions rarely performing above chance level. Intuitively, as each additional corruption makes the image harder to recognize (see Figure 1), it makes sense that this pattern emerges. Perhaps more surprisingly, both methods achieve

| Number of Corruptions | | 1 | 2 | 3 | 4 | 5 | 6 |
|---|---|---|---|---|---|---|---|
| EMNIST | ERM | 88.2 (0.1) | 79.5 (0.1) | 17.6 (0.1) | 7.4 (0.3) | 2.8 (0.2) | 2.5 (0.1) |
| | Contrastive | 87.7 (0.1) | 79.7 (0.2) | 19.5 (0.4) | 8.6 (0.4) | 3.1 (0.2) | **2.9** (0.2) |
| | Modular | **88.8** (0.0) | **85.1** (0.5) | **54.2** (3.2) | **12.9** (1.6) | **3.4** (0.3) | 2.3 (0.1) |
| CIFAR-10 | ERM | 88.5 (0.2) | 66.4 (0.6) | 26.1 (1.2) | 13.5 (0.2) | 10.2 (0.1) | 9.6 (0.5) |
| | Contrastive | 89.2 (0.2) | 67.6 (1.8) | 28.6 (2.1) | 14.7 (0.8) | 9.9 (0.3) | 9.2 (0.2) |
| | Modular | **89.4** (0.1) | **78.5** (0.9) | **60.5** (6.1) | **23.9** (5.8) | **12.3** (1.2) | **10.4** (0.4) |
| FACESCRUB | ERM | 95.1 (0.1) | 90.1 (0.3) | 35.4 (4) | 2.3 (0.3) | 0.5 (0.1) | 0.3 (0.0) |
| | Contrastive | **96.4** (0.2) | **92.1** (0.5) | 38.9 (6.8) | 1.6 (0.2) | 0.4 (0.0) | 0.3 (0.0) |
| | Modular | 95.8 (0.2) | 89.9 (1.0) | **48.8** (25.1) | **5.3** (2.7) | **1.1** (0.4) | **0.4** (0.1) |

Table 1: Compositional robustness shown for compositions of different numbers of corruptions. The median accuracy over all compositions for a given number of corruptions is calculated, this table reports the maximum median (and standard deviation) over three random runs.

accuracy far above chance for compositions of 2 corruptions and perform relatively well for compositions of 3 corruptions despite these domains being outside of the training distribution. We also see that the contrastive training approach makes only minor improvements over ERM, with the most improvement for CIFAR-10. This runs counter to our assumption that encouraging invariance amongst training domains would increase compositional robustness. Finally, we note that neither method optimally solves the task, some compositions of 2 corruptions contain only invertible corruptions, yet neither method reaches ceiling performance for any composition of 2 corruptions.

## 4.2 The Modular Approach Achieves the Best Compositional Robustness

Comparing all three training approaches, we observe that the modular approach outperforms both ERM and contrastive training, with higher average performance in almost all cases in Figure 2. The only exception is on compositions of 2 corruptions for FACESCRUB, where the modular approach is marginally outperformed by contrastive training. We can observe that, in general, explicitly modularizing the modelling of elemental corruptions outperforms the direct encouragement of invariance in terms of compositional robustness. This demonstrates that the monolithic approaches are unable to learn to modularize the structure of the task in the same way as the modular approach. Figure 2 additionally shows the spread of results over different numbers of corruptions. Across methods there are always some compositions of corruptions that are much harder to resolve than others. For example, for compositions of two corruptions using CIFAR-10, there are several compositions which achieve accuracy of 70-90%, but several others below 30%. Overall, no method is yet fully robust to compositions of corruptions.

A tabular view on the performance and variability over seeds is provided in Table 1, where we see the modular approach outperforming alternative methods in almost all cases as in Figure 2. Table 1, reports the maximum median performance over seeds along with the standard deviation. This metric is chosen because, within a fixed number of corruptions, most of the variance for the modular approach comes from the placement of the modules rather than the effect of modularizing visual processing. In these results we aim to explore primarily the effect of good modularization when compared with other methods for improving compositional robustness, the effect of module location on results is investigated further in Section 4.4.

## 4.3 In-Distribution Invariance Does Not Correlate With Compositional Robustness

To investigate our findings further, we examine the invariance scores for the different approaches splitting test domains by the number of elemental corruptions they include to plot correlations. In Figure 3, top row, we plot the elemental invariance score against accuracy on compositions for EMNIST. We observe no meaningful correlation between elemental invariance scores and accuracies on compositional test domains, with high p-values and low r-values. This runs counter to our initial expectations based on the ubiquity of invariant representation learning in the domain generalization literature. For our task, these results indicate that encouraging invariance between representations on the training domains may be insufficient to achieve

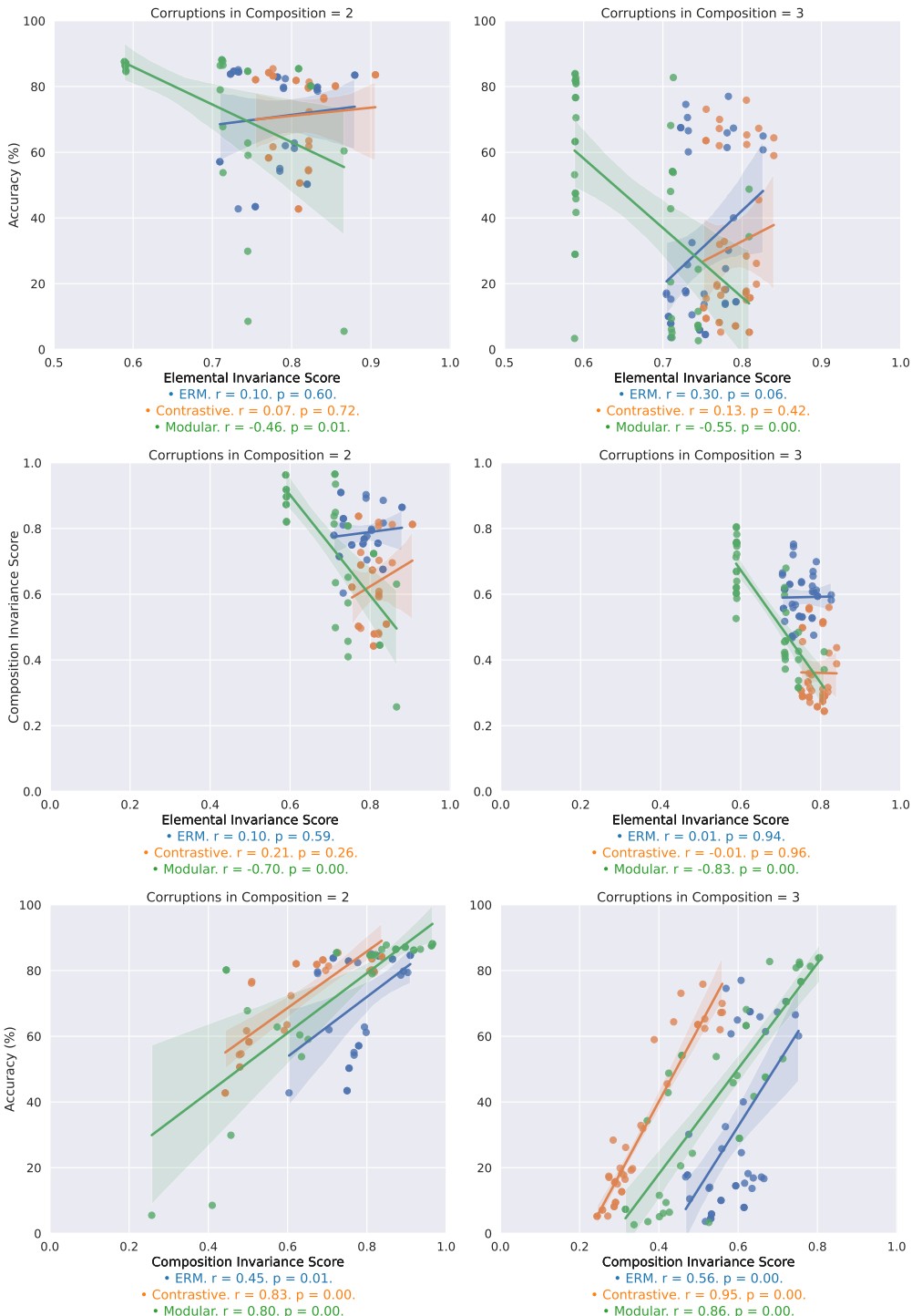

Figure 3: Correlating invariance scores with compositional robustness for EMNIST. Row one shows the level of representational invariance amongst elemental corruptions fails to correlate with compositional robustness (accuracy). Row two shows the lack of dissemination of invariance between elemental corruptions and compositions. Row three plots composition invariance scores against compositional robustness. Columns show subsets of evaluation domains depending on the number of elemental corruptions making up a composition (as in Figure 2).

robustness. We even see some points for the modular approach (in the upper left of the plots) that achieve higher accuracy than ERM or contrastive training achieve on any domain yet have lower invariance scores.

We also note that contrastive training only slightly increases the observed invariance between elemental corruptions, with a small rightward shift of points when compared to ERM. One possible reason for this smaller than expected increase may be because we set hyper-parameters on the training domains (Gulrajani & Lopez-Paz, 2022) and high contrastive weights take away from in-distribution performance. Alternatively, there has been some discussion on whether the contrastive loss improves performance because of increased invariance or by other mechanisms Shen et al. (2022); Sakai et al. (2022).

Row three of Figure 3 shows strong positive correlations between the composition invariance score and accuracy on compositions. This is as expected, since a high composition invariance score indicates a similar representation between compositions and elemental corruptions (which all achieve good accuracy). However, in row two of Figure 3 we again see limited, or even negative, correlations between elemental and composition invariance scores. This suggests that invariance built on elemental training domains may not transfer to invariance on compositional test domains, so we cannot consistently improve the composition invariance score by encouraging elemental invariance.

By and large these trends are consistent over datasets (Appendix E) and seeds (Appendix F). A notable exception is the negative correlation for the modular approach in row two of Figure 3 is not seen in other datasets. We also observe a positive correlation between elemental invariance score and accuracy for ERM on CIFAR-10. On CIFAR-10, the encouraging of invariance with contrastive training builds slightly more invariant representations but the correlation between elemental invariance and accuracy disappears.

### 4.4 Practical Limitations

The aim of this work is to provide greater understanding of the factors that influence compositional robustness in neural networks. In particular, it is not our aim to provide an directly applicable method for improving compositional robustness. Nevertheless we now show some additional experiments to briefly highlight some of the practical limitations of the modular approach taken in this study.

Firstly, compared to the monolithic approaches, the modular approach has substantially higher variance over seeds (particularly for FACESCRUB in Table 1). This is primarily due to variance in the selection of the module locations. To show this, Table 2 shows results for individual seeds for the modular approach evaluated with FACESCRUB using alternative module locations. In Table 2 for *Seed 1* the automatic module placement strategy described in Section 3.3 finds a placement that outperforms the monolithic approaches (Table 1). However, for the remaining seeds, the automatically placed modules show substantially worse performance, hence why Table 1 shows such high variance for the modular approach. The *Manual* columns of Table 2 show that by manually setting the locations of the modules to match those found for *Seed 1*, these alternate seeds achieve similar accuracy to *Seed 1* demonstrating that most of the variance is due to the difficulty of placing modules in the correct location. This result emphasises the strength of the modular approach for handling compositional structure but highlights the challenge of how to correctly architect modular systems as we cannot feasibly try every module in every location and often we do not have compositional validation data for model selection (Gulrajani & Lopez-Paz, 2022).

Additionally, apart from ERM all of the evaluated methods require paired data between domains which is an unrealistic expectation in practical applications. For modular approaches we must know which corruptions are applied in a given test domain in order to apply the correct modules. Another interesting angle for future, more practically minded, solutions is to remove or reduce these assumptions.

## 5 Discussions

We end with several discussions on different interpretations of this work and links to larger questions that may motivate future work.

**What is the structure of natural data?** In our compositional robustness framework we see only the elemental factors of variation (elemental corruptions) during training. In reality, whilst it is likely not possible to see every composition, most real-world data will contain an unstructured sampling of the compositional

| Num. Corrs. | | 1 | 2 | 3 | 4 | 5 | 6 |
|---|---|---|---|---|---|---|---|
| **Seed 1** | *Automatic* | 95.5 | 87.9 | 48.8 | 5.3 | 1.1 | 0.4 |
| | *Manual* | 95.5 | 87.9 | 48.8 | 5.3 | 1.1 | 0.4 |
| **Seed 2** | *Automatic* | 95.8 | 88.7 | 7.7 | 0.8 | 0.3 | 0.3 |
| | *Manual* | 95.7 | 89.1 | 47.1 | 5.5 | 0.9 | 0.4 |
| **Seed 3** | *Automatic* | 95.6 | 89.9 | 3.3 | 0.4 | 0.3 | 0.3 |
| | *Manual* | 95.4 | 89.4 | 47.9 | 4.6 | 0.6 | 0.4 |
| **Mean** | *Automatic* | **95.6** (0.2) | **88.8** (1.0) | 19.9 (25.1) | 2.2 (2.7) | 0.6 (0.5) | 0.3 (0.1) |
| | *Manual* | 95.5 (0.2) | **88.8** (0.8) | **47.9** (0.9) | **5.1** (0.5) | **0.9** (0.3) | **0.4** (0.0) |

Table 2: Comparing compositional robustness for FACESCRUB using automatic module placement vs. manually placing modules at the locations found when using automatic placement for *Seed 1*. The median accuracy over all compositions for a given number of corruptions is calculated. The final row reports the mean (and standard deviation) of these median values over the three random runs.

space. This assumes however, that it is possible to decompose data from the environment into elemental factors of variation (Chen et al., 2016; Higgins et al., 2017; Kim & Mnih, 2018; Roth et al., 2023) or independent (causal) mechanisms (Peters et al., 2017; Schölkopf et al., 2021; Parascandolo et al., 2018). At present it remains unknown if there exists a practically sized set of elemental transformations from which all visual stimuli can be composed, but if such a set exists, the ideas presented in this work suggest that modular architectures may be able to model this space more efficiently than large monolithic models.

**Learning to decompose from data.** If there exists a set of elemental transformations from which all visual stimuli can be composed, and we are to make use of modularity as an inductive bias to model them, we must *learn* how to decompose datasets into their constituent factors and how to modularize knowledge in the appropriate semantic spaces (Goyal & Bengio, 2022). In this work we have shown that modular approaches have the potential to surpass previous approaches if the decomposition is available and progress has been made on finding appropriate semantic spaces (Royer & Lampert, 2020; Eastwood et al., 2022a). The learning of decompositions remains an open problem (Parascandolo et al., 2018; Bengio et al., 2019; Locatello et al., 2019; Goyal et al., 2021).

**How modular should neural networks be?** The modular approach taken in this study uses neural network layers as modules which are manually assigned to handle specific corruptions, yet we have also experimented with monolithic networks and with using entirely separate networks for each corruption (Section 4.4). Even if we are able to decompose data into constituent factors, there remains a question of what degree of modularity should be used to model these factors. There have been recent exciting empirical studies in this direction (D'Amario et al., 2021; Madan et al., 2022; Yamada et al., 2023) but no consensus has yet been reached.

**What is a module and why do they work?** We consider modular methods to be exactly those methods that explicitly parameterize domain specific information. For example, rather than only encouraging invariance across elemental corruptions we could additionally have provided each sample with a label for each corruption. Adding a label can be seen as a special, less general, case of modularity. In the case of a one hot label, this is equivalent to adding a learnable bias vector per-corruption in the first layer of the network. This means a set of corruption specific parameters is added per-corruption, i.e. that we can consider this a module.

In this work, we use this understanding of modularity to broadly categorize approaches for improved robustness and domain generalization into three main strands. Firstly, *ERM*, where we train jointly with no task-specific inductive biases. Secondly, *invariance*, where we try to build invariant representations without including any domain specific information. Thirdly, *modularity*, which is any method that includes separate processing of domain information, explicitly (as we do) or implicitly by encouraging modules to form or by adding labels. In this work we find modular networks often outperform monolithic networks that do not contain any domain specific information. It may even be possible to get the benefits of different approaches, with recent works beginning to investigate hybrid solutions that make use of both invariance and domain specific information (Eastwood et al., 2023).

Our results, and the results of others (Zhao et al., 2019; Johansson et al., 2019; Rosenfeld et al., 2021; Shen et al., 2022), raise questions about whether encouraging invariance alone is sufficient to achieve domain generalization in general. We know that invariance is a key factor for robust generalization but we do not yet know how far invariance will be able to take us. Perhaps we simply need to better understand and implement the neural mechanisms that allow invariances to build (Anselmi et al., 2016; Poggio & Anselmi, 2016; Schölkopf et al., 2021; Goyal & Bengio, 2022), or we may need to further explore learning representations that are only partially invariant (Kong et al., 2022; Shen et al., 2022; Sun et al., 2023).

## 6 Conclusion

Since the visual space containing all corruptions is compositional in nature, we have introduced a new framework to evaluate the compositional robustness of different models. We have observed that modular approaches outperform monolithic approaches on this task, even when invariant representations are encouraged. For domain generalization tasks with compositional structure our results raise questions about the efficacy of encouraging invariance without further inductive biases. This work represents only a first step in understanding how neural networks behave under compositional structures, further research is needed into developing methods that make fewer assumptions about the information available at test time and that can work with large unstructured datasets where factors of variation are unknown.

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

# Appendix

## Table of Contents

# A  Choice of Corruptions

The choice of corruptions used in our compositional robustness task is quite subtle. We want to ensure a good mixture of different types of corruptions and the compositions they form, but without creating a compositional space that is so big that it becomes prohibitively expensive to evaluate. Due to the exponential increase in the number of possible compositions as the number of elemental corruptions increases, and in order to reduce computational costs, we make the following concessions: (i) we keep the total number of elemental corruptions low whilst ensuring a good mixture of elemental corruptions; (ii) we include compositions constructed from every combination of elemental corruptions but sample the possible permutations (orderings) of elemental corruptions that make up a composition (see Appendix A.1); (iii) we do not consider the 3D projection problem (see Appendix A.3).

As discussed in Section 3.1, along with the *Identity (ID)* data we consider the corruptions, *Contrast (CO)*, *Gaussian Blur (GB)*, *Impulse Noise (IM)*, *Invert (IN)*, *Rotate 90° (R90)* and *Swirl (SW)*, which can be seen for EMNIST and CIFAR-10 in Figures 4 and 5 respectively. We consider two different behaviors that corrupting functions may exhibit and select this set of corruptions to get a mixture of behaviors. Firstly, corruptions can be local or long-ranged, where images under local corruptions (such as *Invert*) can be transformed to the *Identity* image by applying a patch-wise operation. On the other hand, long-ranged corruptions (such as *Rotate 90°*) require a holistic understanding of the image. Additionally, corruptions can be lossless or lossy, where lossless corruptions lose no information so can be perfectly inverted and lossy corruptions may lose information due to randomness or the application of non-invertible corrupting functions.

## A.1  Sampling and Commutativity

Our set of elemental corruptions allows us to consider compositions made up of up to six corruptions at once (we do not allow for repeated application of elemental corruptions). As not all elemental corruptions are commutative under composition (e.g. $IM \circ GB \neq GB \circ IM$), we must take into account the possible orderings of elemental corruptions when constructing compositions. When taking into account possible orderings there are $^6P_2 = 30$ possible orderings of two corruptions but $^6P_6 = 720$ possible ordering of six corruptions, where $^nP_r = n!/(n-r)!$, counts the number of possible permutations. As we don't want results to be dominated by compositions of larger numbers of corruptions and to reduce the number of compositional test domains, we *sample* the possible orderings.

For compositions of two corruptions, we consider all possible orderings giving $^6P_2 = 30$ compositions. For compositions of more than two corruptions we aim to get as close to $^6P_2 = 30$ test domains as possible whilst maintaining a balance of the possible unique combinations of elemental corruptions. This means we first calculate the number of unique combinations as $^nC_r = n!/r!(n-r)!$, where $n$ is the total number of elemental corruptions and $r$ is the number of elemental corruptions in the compositions we are considering. We then sample the same number of possible orderings of each unique combination until we get as close as possible

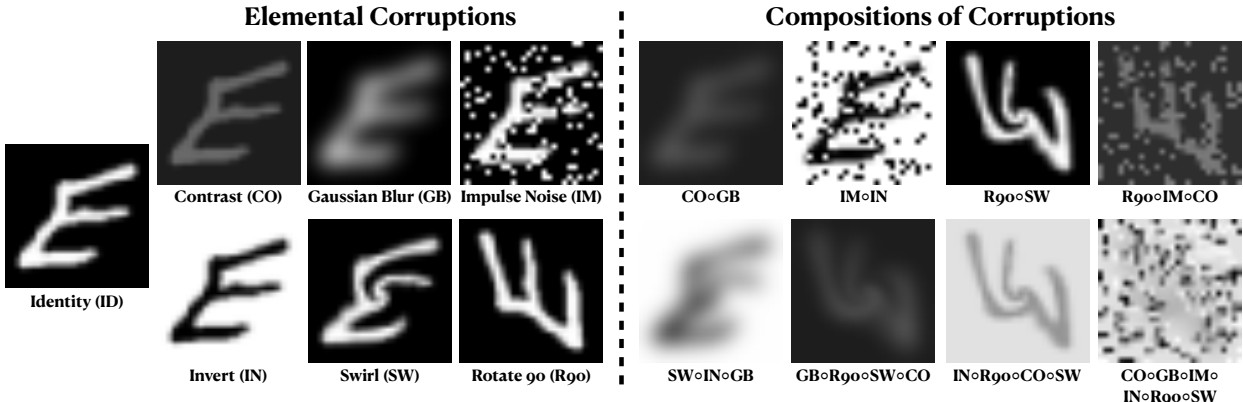

Figure 4: The compositional robustness task for EMNIST.

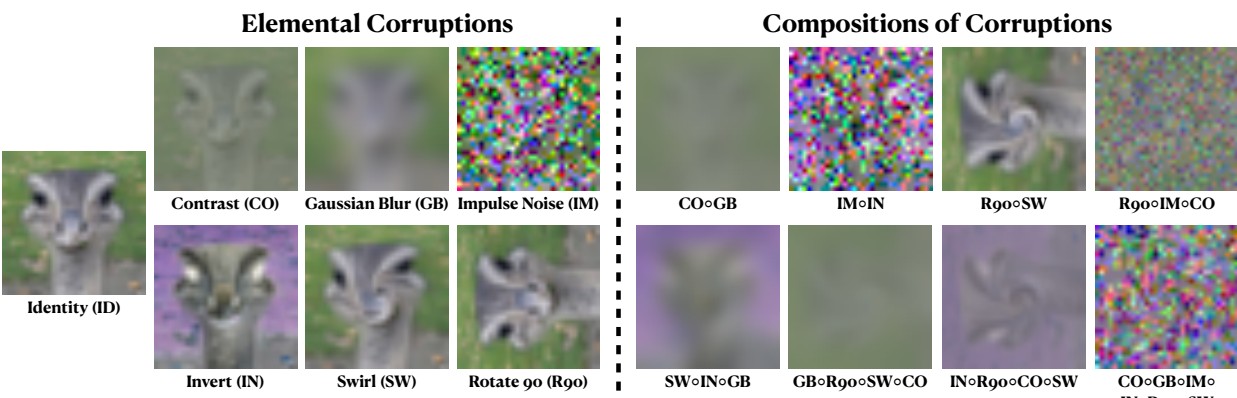

Figure 5: The compositional robustness task for CIFAR-10.

to 30 domains. As an example, for compositions of three corruptions $^{6}C_3 = 20$, so we have twenty unique combinations of three elemental corruptions. For each unique combination we sample two possible orderings, giving forty test domains. For compositions of four corruptions we have fifteen unique combinations so we again sample two possible orderings, for compositions of five corruptions we have six unique combinations so we sample five orderings and for compositions of six corruptions there is only one unique combinations so we sample thirty different orderings.

## A.2 Corruption Severity

Our implementation allows for corruptions to be applied with differing severity, for example by adding more or fewer random pixels for *Impulse Noise* or by increasing or decreasing the Gaussian filter size when creating *Gaussian Blur*. In our main experiments we keep the severity fixed as varying the severity would significantly increase the size of the compositional space. To show the impact of varying severity we perform one additional ablation on EMNIST using the lowest severity settings in Figure 6, where we verify that the relative performance of different approaches remains unchanged.

For corruptions with variable severity, the specific functions we use in our experiments are as follows: *Contrast* calculates per-channel $x * s + (1 - s)\bar{x}$ for pixel value $x$ and average pixel value $\bar{x}$ using $s = 0.2$; *Gaussian Blur* uses a Gaussian filter with standard deviation 2; *Impulse Noise* uses random salt and pepper noise affecting 27% of the pixels in an image; and *Swirl* uses the scikit-image (van der Walt et al., 2014) implementation of image swirling with strength 3 and radius equal to the image width in pixels divided by $\sqrt{2}$. Whilst the scale of severity is somewhat arbitrary, following the definitions of MNIST-C (Mu & Gilmer, 2019), we use severities 2, 2, 5 for *Contrast*, *Gaussian Blur* and *Impulse Noise*; *Swirl* uses an original implementation with severity 3 according to our severity scale.

## A.3 The 3D Projection Problem

A final point of interest when choosing which elemental corruptions to consider is the problem of 3D projection. There are certain corruptions that occur in natural data that are inherently 3-dimensional, yet we only see the results as a projection onto a 2-dimensional image plane. This fact introduces complexity in the way corruptions can be applied and composed if we are aiming to create a system with vision that is as robust as humans.

To see the problem, consider the corruption *Scale (SC)*, where we create a zoomed out version of a base image (see Figure 7). Imagine that we then also consider the composition of *Scale* with *Gaussian Blur*. *SC∘GB* creates a very different image to *GB∘SC*, but more importantly these represent fundamentally different processes in the 3D world. If scaling is applied before blurring this corresponds to the case where there is a fixed amount of blur in the scene (e.g. because of an eye condition) and the object we care about is moved further away from the viewer. On the other hand if blurring is applied before scaling this corresponds to the

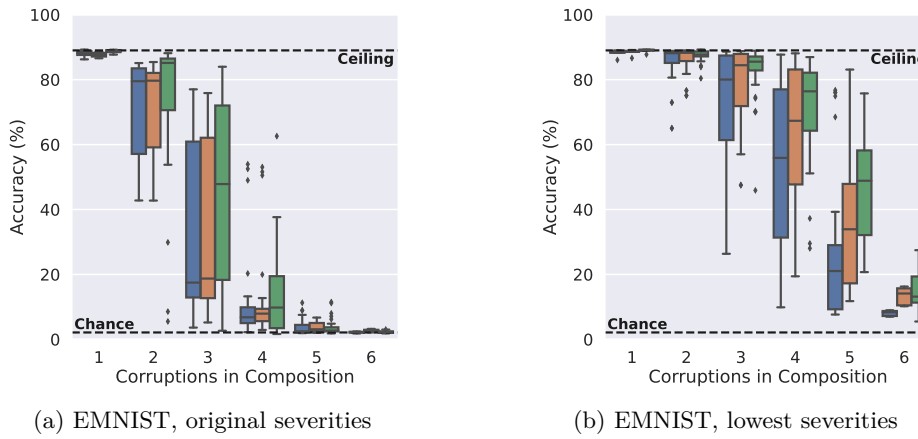

(a) EMNIST, original severities          (b) EMNIST, lowest severities

Figure 6: Comparing performance of different methods when using different corruption severities. (a) is the severities used in the main text in Figure 2a, (b) is using the lowest severity setting for each corruption. With lower severities, performance is higher for all methods but relative performance remains the same

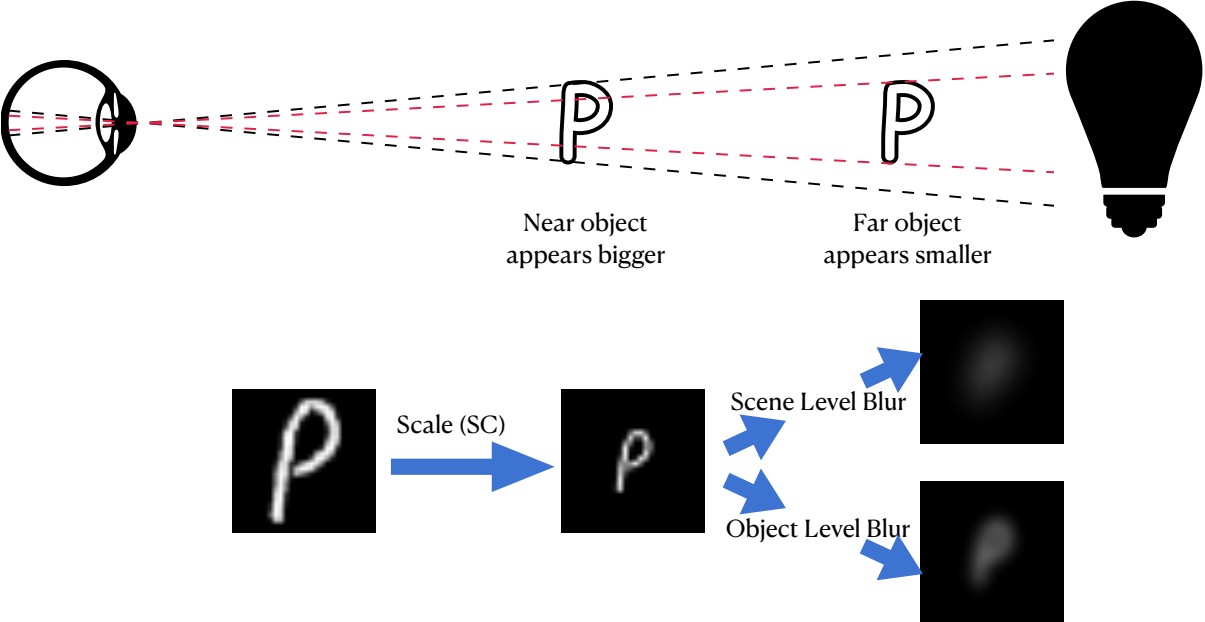

Figure 7: The 3D projection problem. Corruptions of 2D images can represent 3D processes, for example scaling an image represents moving the object further away from the viewer (top). When composing corruptions, this can lead to different orderings of corruptions representing different 3D processes (bottom).

case where the object itself is blurry (e.g. because of damage around the edges). This process is depicted in Figure 7.

The point of this discussion is to demonstrate that applying a corruption at the scene level can be fundamentally different from applying a corruption at the object level. Whilst this can be taken into account (e.g. by changing the order of *Gaussian Blur* and *Scale*), we aim to avoid this situation by only considering corruptions where changing the ordering under composition does not change the composition from a scene level process to an object level process (or vice versa). This makes our task more practical as we can apply it to any image classification dataset. Since we may not even consciously perceive the effect of scaling accurately (Sperandio & Chouinard, 2015; Köhler, 1970), future work may find that different processes in 3D space should be handled in different ways or at different levels of abstraction.

# B    Module Implementation and Interpretability

Figure 8 shows the training process for a module trained on the *Invert* corruption. First a network is trained on *Identity* data to learn parameters $\boldsymbol{\theta}_{shared}$. These weights are then frozen (gray boxes in Figure 8) and a module is trained to 'undo' the *Invert* corruption in latent space (blue box in Figure 8). To train the module, the contrastive loss is used to align representations of *Identity* data before the module is applied with representations of *Invert* data after the module is applied. As described in the main text, we also use the cross entropy loss to ensure classification accuracy is maintained.

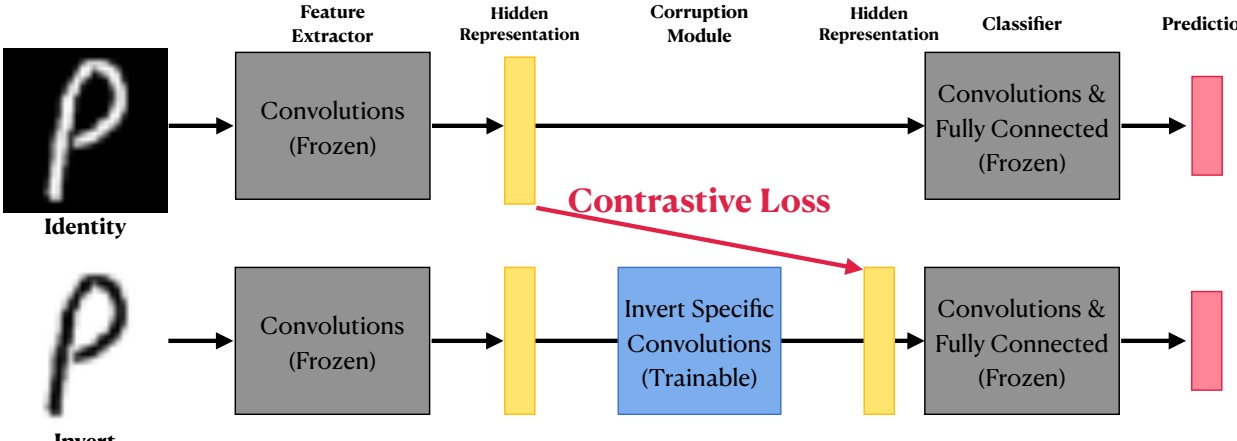

Figure 8: Module training diagram. After pre-training on *Identity* data, shared network parameters are frozen (gray boxes) and a module (blue box) is trained to align the representation of the corrupted *Invert* image with the representation of the *Identity* image using the contrastive loss. In this figure, apart from those of the module, all parameters are identical between the top and bottom networks.

Using interpretability tool Deephys (Sarkar et al., 2023), we visualize the effect of modules trained in this way in Figure 9 . We find neurons which are initially activated by very different class instances when comparing *Identity* data with corrupted data, but after applying the module, neurons fire for similar class instances between the *Identity* and corrupted data.

| Pre-Module | Post-Module | Identity Data |
|---|---|---|

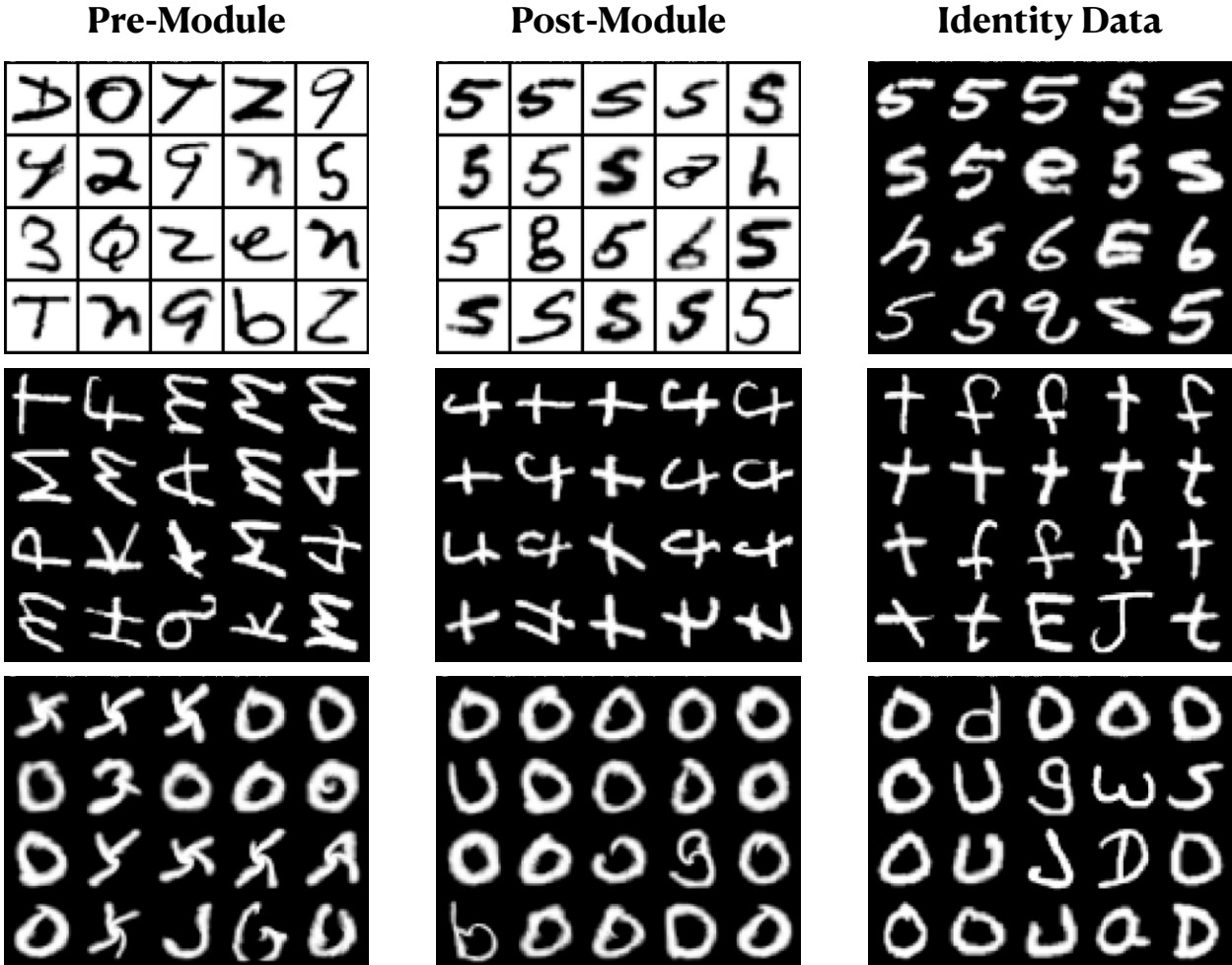

Figure 9: Training modules to undo elemental corruptions. Images in this grid represent some of the images that maximally activate a neuron. The first column shows a neuron before the module is applied, this is equivalent to the images that the neuron is tuned to for a network trained only on *Identity* data. The second column shows the same neuron after the module is applied. The third column shows how the *Identity* data activates this neuron. By comparing across columns we can see that modules learns to align the hidden representations so that neurons fire for similar class instances between *Identity* and corrupted data. Top to bottom the corruptions in the rows are, *Invert*, *Rotate 90°* and *Swirl*.

## C Elemental and Composition Invariance Scores, a Worked Example

This section gives an exemplar activation grid and a worked example of calculating elemental and composition invariance scores based on the techniques described by Madan et al. (2022).

Table 3 shows an exemplar activation grid for a single neuron for the test domain containing the composition $CO \circ GB$ on CIFAR-10. We see the 4 rows consist of the composition alongside the elemental corruptions that are relevant for $CO \circ GB$, and the 10 columns for each of the 10 categories of CIFAR-10, creating domain-category pairs.

To calculate the invariance scores for this example we first find the preferred category as $j^* = \arg\max_j \sum_{i=1}^3 a_{i,j}$, which indicates that this neuron activates maximally for category 7. The elemental invariance score is the worst case difference amongst the elemental corruption activations for this category (the maximum is marked $*$, and the minimum is marked $\dagger$).

Table 3: An exemplar activation grid

|  | Cat. 1 | Cat. 2 | Cat. 3 | Cat. 4 | Cat. 5 | Cat. 6 | Cat. 7 | Cat. 8 | Cat. 9 | Cat. 10 |
|---|---|---|---|---|---|---|---|---|---|---|
| *CO* | 0.002 | 0.007 | 0.038 | 0.089 | 0.039 | 0.794 | 0.998 | 0.015 | 0.022 | 0.005 |
| *GB* | 0.011 | 0.021 | 0.070 | 0.144 | 0.061 | 0.733 | 0.955† | 0.043 | 0.029 | 0.020 |
| *ID* | 0.020 | 0.004 | 0.051 | 0.090 | 0.039 | 0.791 | 1.000* | 0.016 | 0.018 | 0.000 |
| *CO∘GB* | 0.035 | 0.102 | 0.109 | 0.126 | 0.087 | 0.415 | 0.638‡ | 0.078 | 0.116 | 0.138 |

$$I_e = 1 - (\max_i a_{i,j^*} - \min_i a_{i,j^*})$$
$$= 1 - (1.000 - 0.955)$$
$$= 0.955$$

The composition invariance score finds the activation amongst the elemental corruptions that is closest to the composition's activation (marked ‡) for the preferred category.

$$I_c = 1 - \min\{|a_{i,j^*} - a_{E+1,j*}|\}_{i=1}^{E}$$
$$= 1 - \min\{|0.998 - 0.638|, |0.995 - 0.638|, |1.000 - 0.638|\}$$
$$= 0.683$$

For this particular neuron, we would deduce that the elemental corruptions have relatively invariant activations whereas the activations are less invariant when we include the composition.

## D  Network Architecture Details

This appendix gives the specific architecture of the simple convolutional network used for EMNIST experiments in Table 4. For CIFAR-10 we use ResNet18 and for FACESCRUB Inception-v3. In both cases we use the official PyTorch (Paszke et al., 2019) implementations of the architectures. Rather than giving a lengthy description of the possible architectures for modules between every layer of these networks we refer the reader to the associated code repository (file lib/networks.py). The architectures of the auto-encoders used in Section 4.4 can also be found in this file.

Table 4: The network architecture used for EMNIST experiments. For convolutions, the weights-shape is: *number of input channels × number of output channels × filter height × filter width.*

| Block | Weights-Shape | Stride | Padding | Activation | Dropout Prob. |
|---|---|---|---|---|---|
| Convolution | $3 \times 64 \times 5 \times 5$ | 2 | 2 | ReLU | 0.1 |
| Convolution | $64 \times 128 \times 5 \times 5$ | 2 | 2 | ReLU | 0.3 |
| Convolution | $128 \times 256 \times 5 \times 5$ | 2 | 2 | ReLU | 0.5 |
| Convolution | $256 \times 256 \times 5 \times 5$ | 2 | 2 | ReLU | 0.5 |
| Linear | $1024 \times 512$ | N/A | N/A | ReLU | 0.5 |
| Linear | $512 \times$ Number of Classes | N/A | N/A | Softmax | 0 |

## E  Invariance Scores for All Datasets

Appendices E, F and G show a large number of plots over the following pages. This appendix contains further plots correlating invariance scores with compositional robustness. To begin we show the invariance summary plots for CIFAR-10 (Figure 10) and FACESCRUB (Figure 11). These plots are the equivalent of Figure 3 for EMNIST from the main text.

Following this, in Figures 12-20, we show the invariance summary plots (Figures 3, 10, 11) expanded over all compositional test domains. That is, these plots include the plots for compositions containing more than three corruptions. For compositions of more than three corruptions accuracy is often low, making it challenging to uncover meaningful trends.

## F  Variance Over Seeds

This appendix shows the figures included in the main text for two further random seeds. In particular we replicate Figures 2, 3, 10 and 11 in each case. The results in the main text come from the first random seed, Figures 21-24 show the second random seed and Figures 25-28 show the third random seed.

## G  Heat Maps - Full Granular Results

Finally we show granular results, showing the individual accuracy for every elemental corruption and composition. This is the raw data that is summarized by the box plots in Figures 2, 21 and 25. We show heat maps for every dataset and every seed in Figures 29-37 to give a per-domain view of the differences in behaviors for the different methods for compositional robustness.

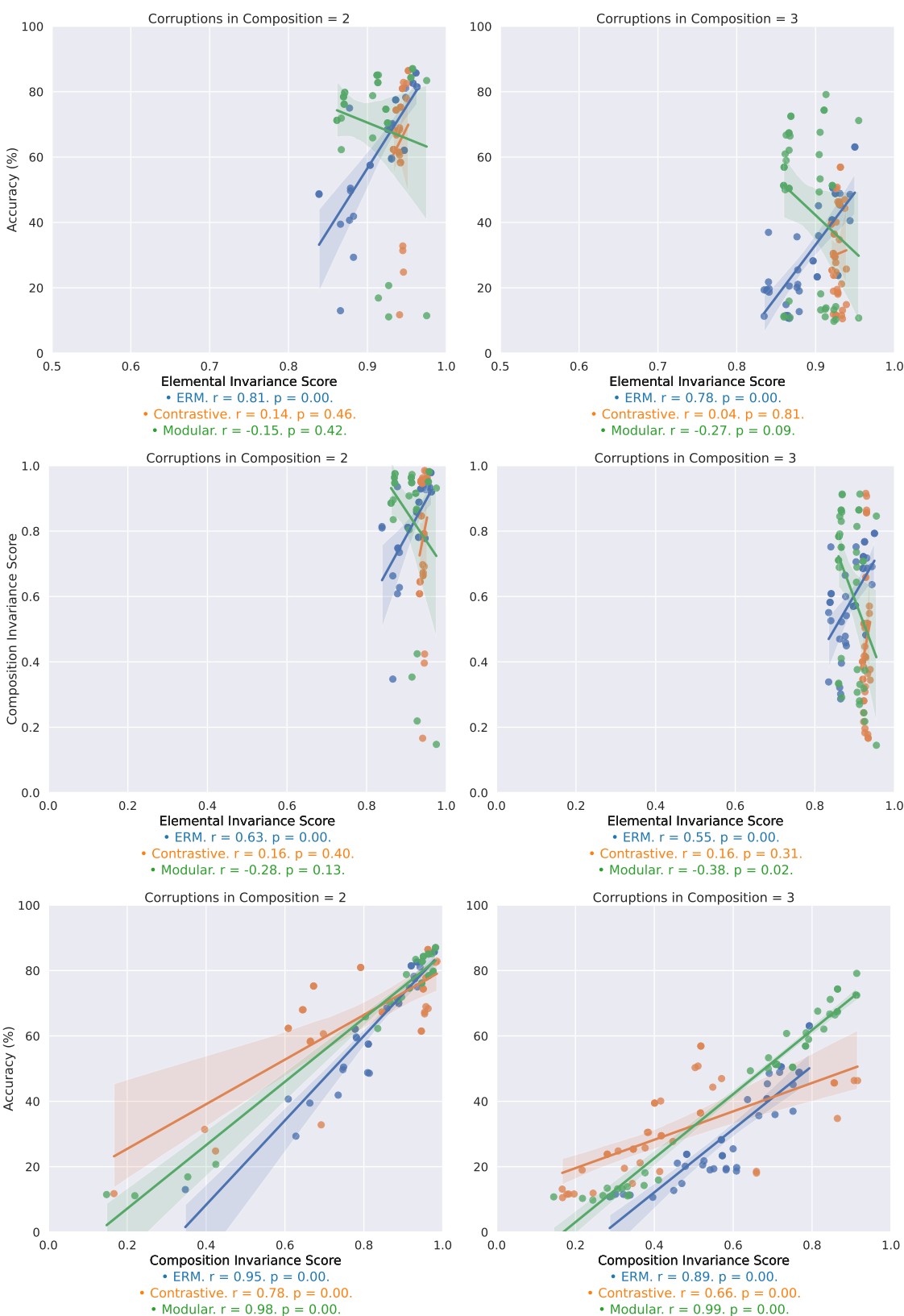

Figure 10: Correlating invariance scores with compositional robustness for CIFAR-10. These plots plot the same relationships as in Figure 3.

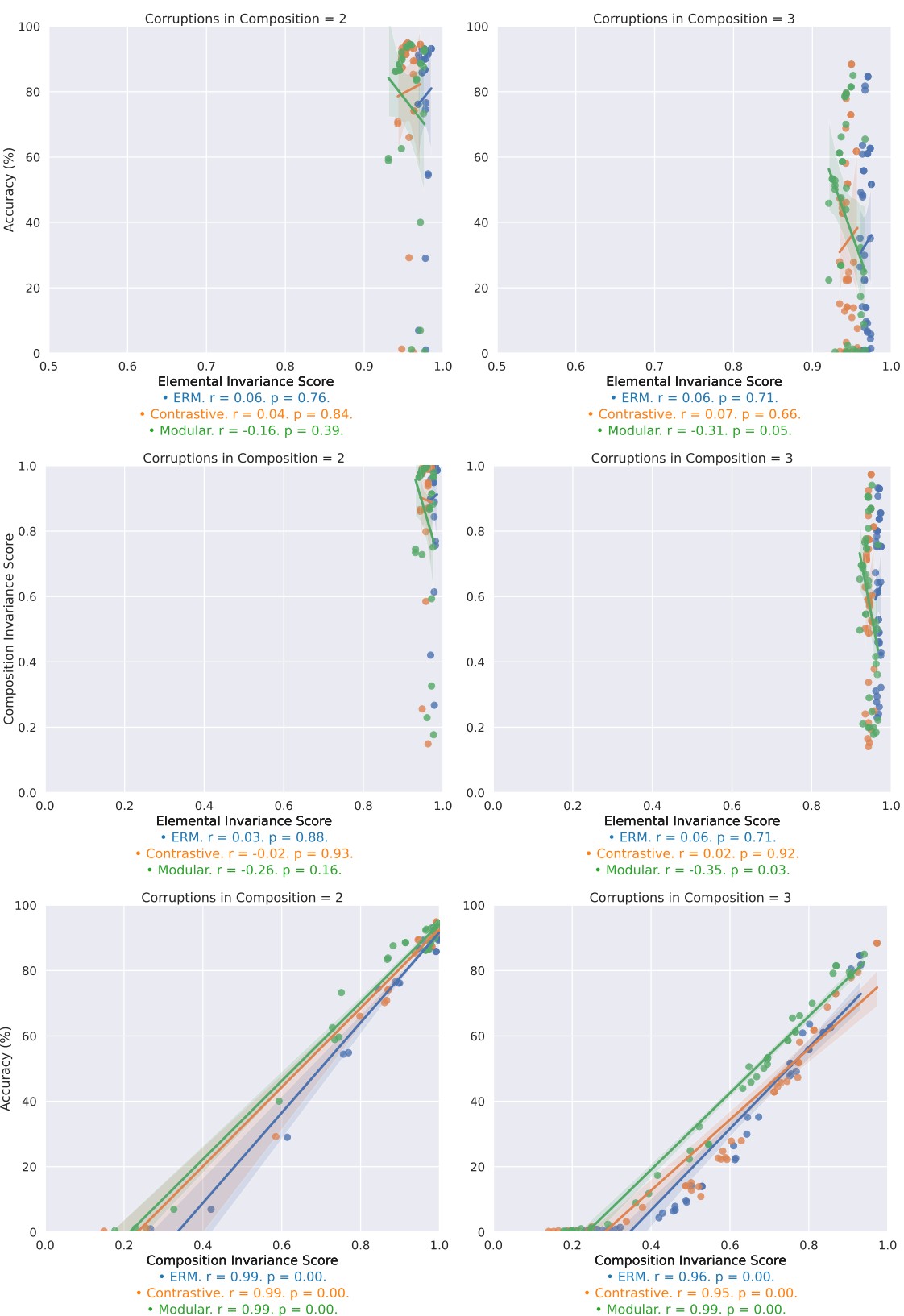

Figure 11: Correlating invariance scores with compositional robustness for FACESCRUB. These plots plot the same relationships as in Figure 3.

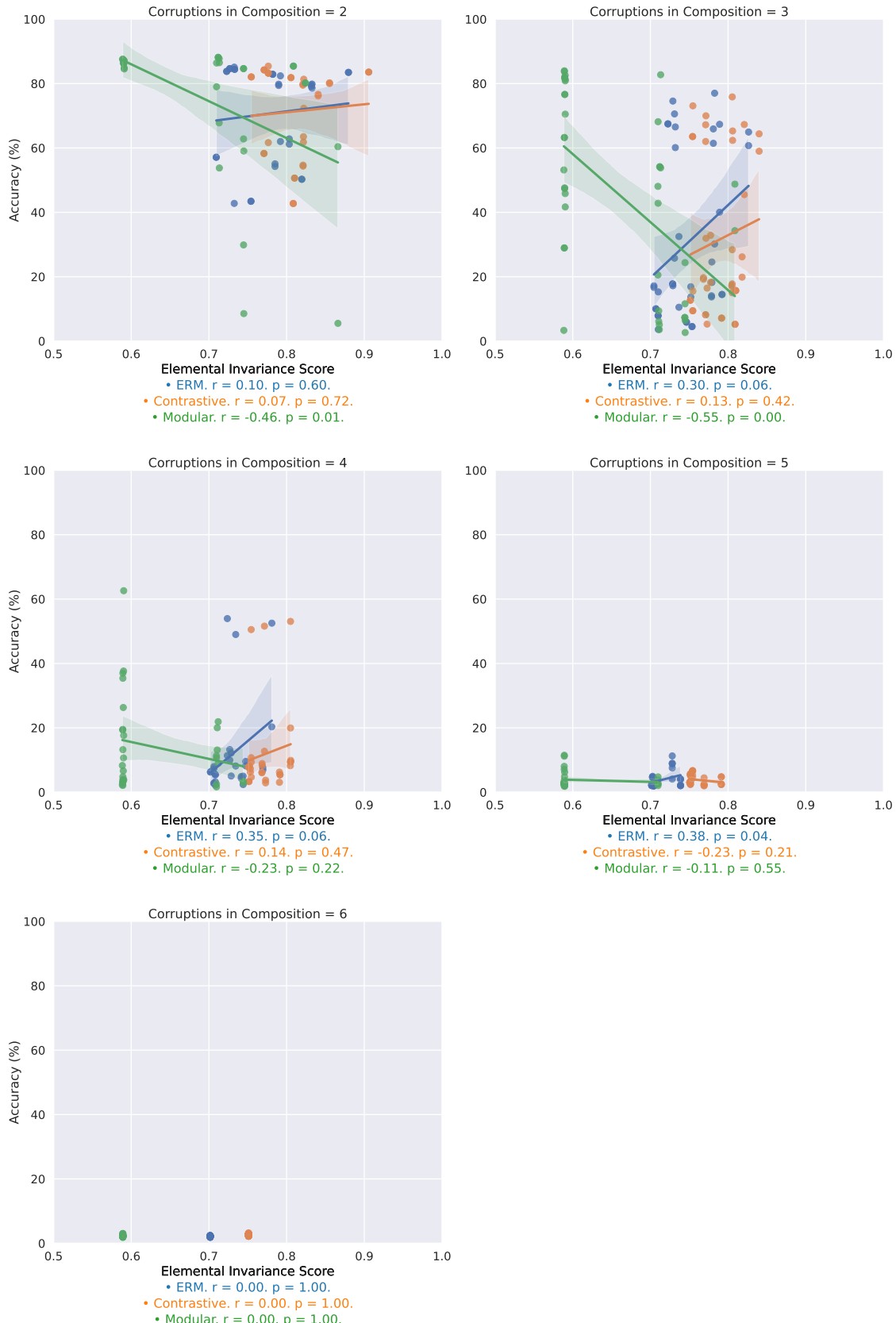

Figure 12: Correlating the elemental invariance score with compositional robustness for EMNIST. These plots expand the first row of Figure 3 to show all compositional test domains.

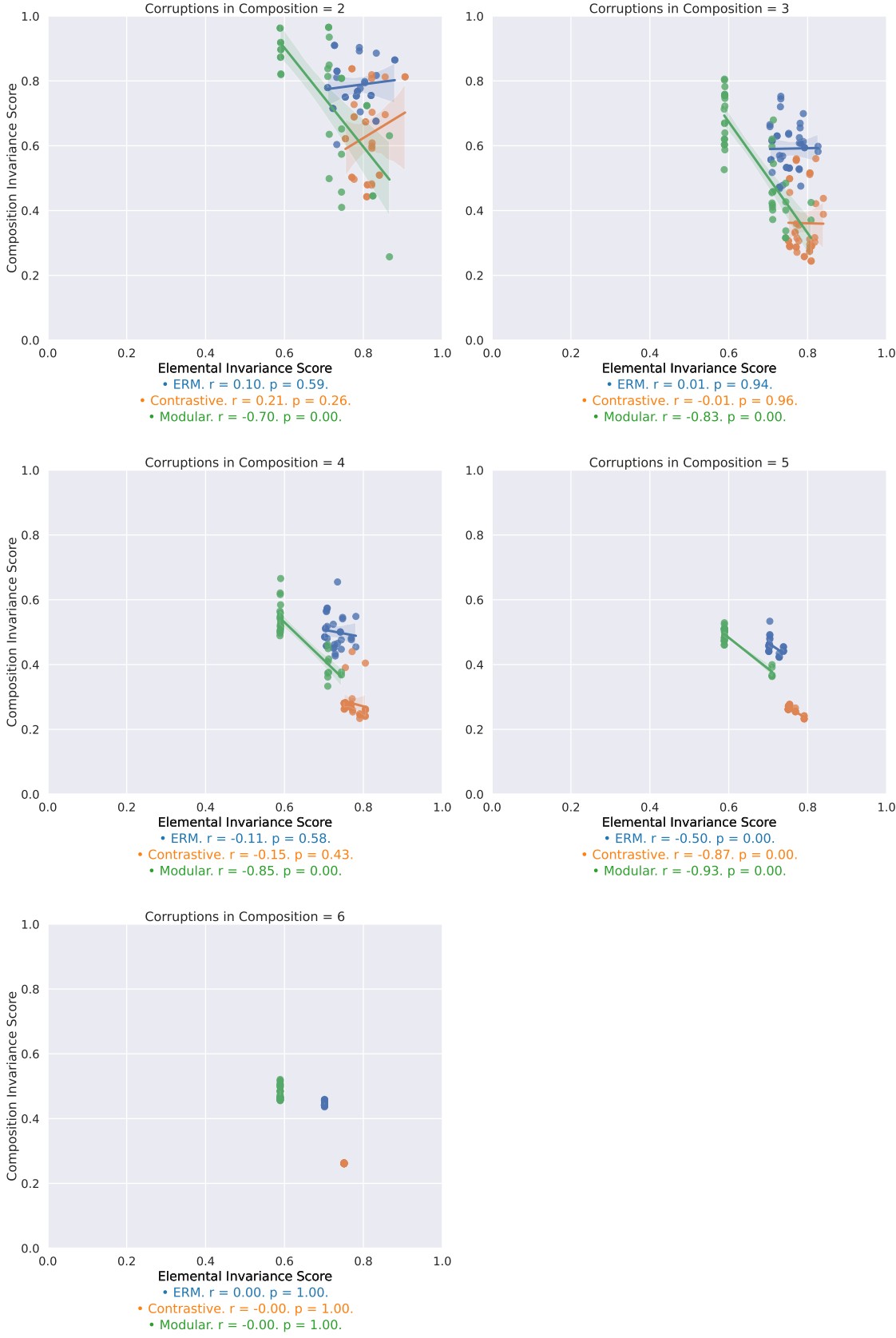

Figure 13: Correlating the elemental invariance score with the composition invariance score for EMNIST. These plots expand the second row of Figure 3 to show all compositional test domains.

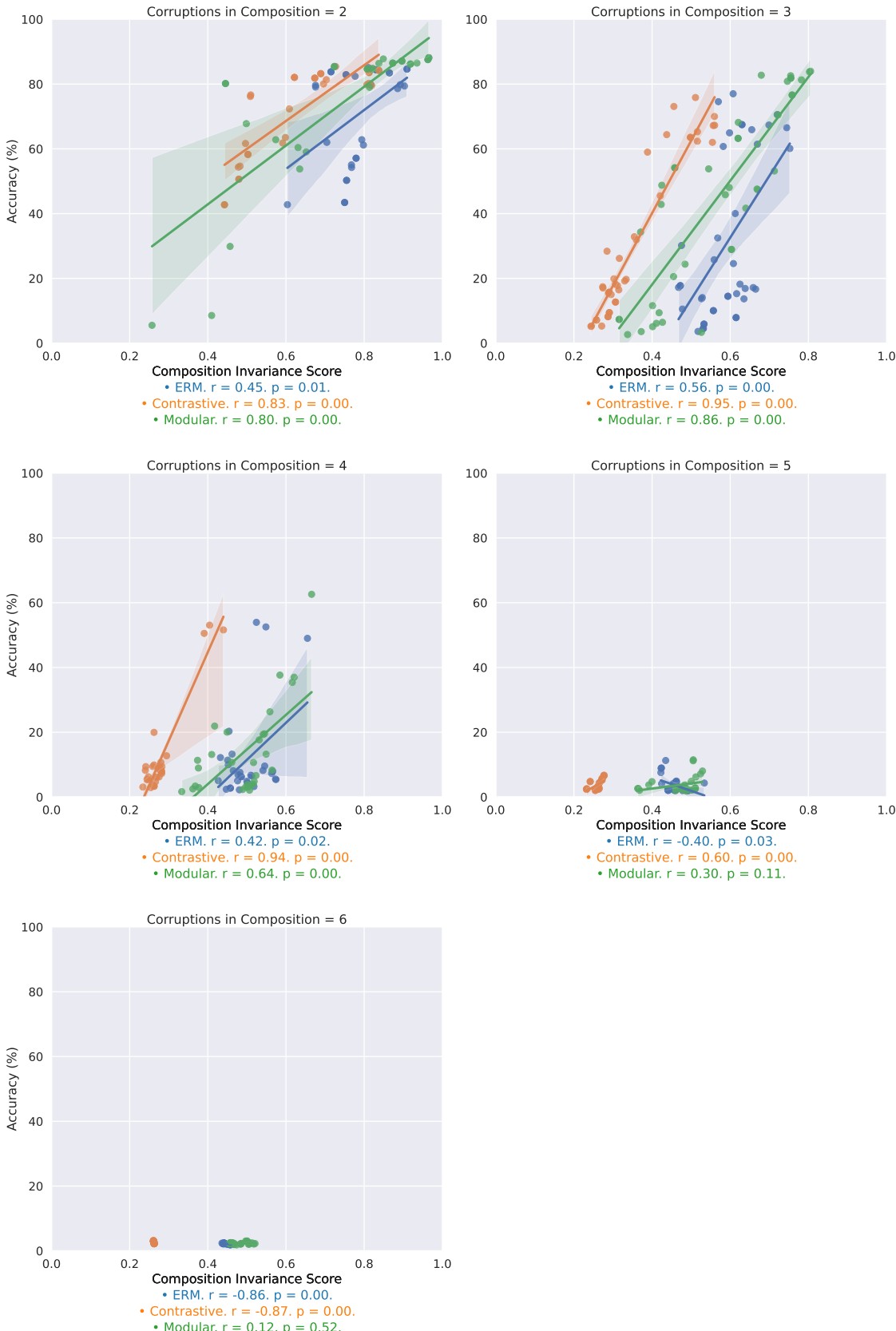

Figure 14: Correlating the composition invariance score with compositional robustness for EMNIST. These plots expand the third row of Figure 3 to show all compositional test domains.

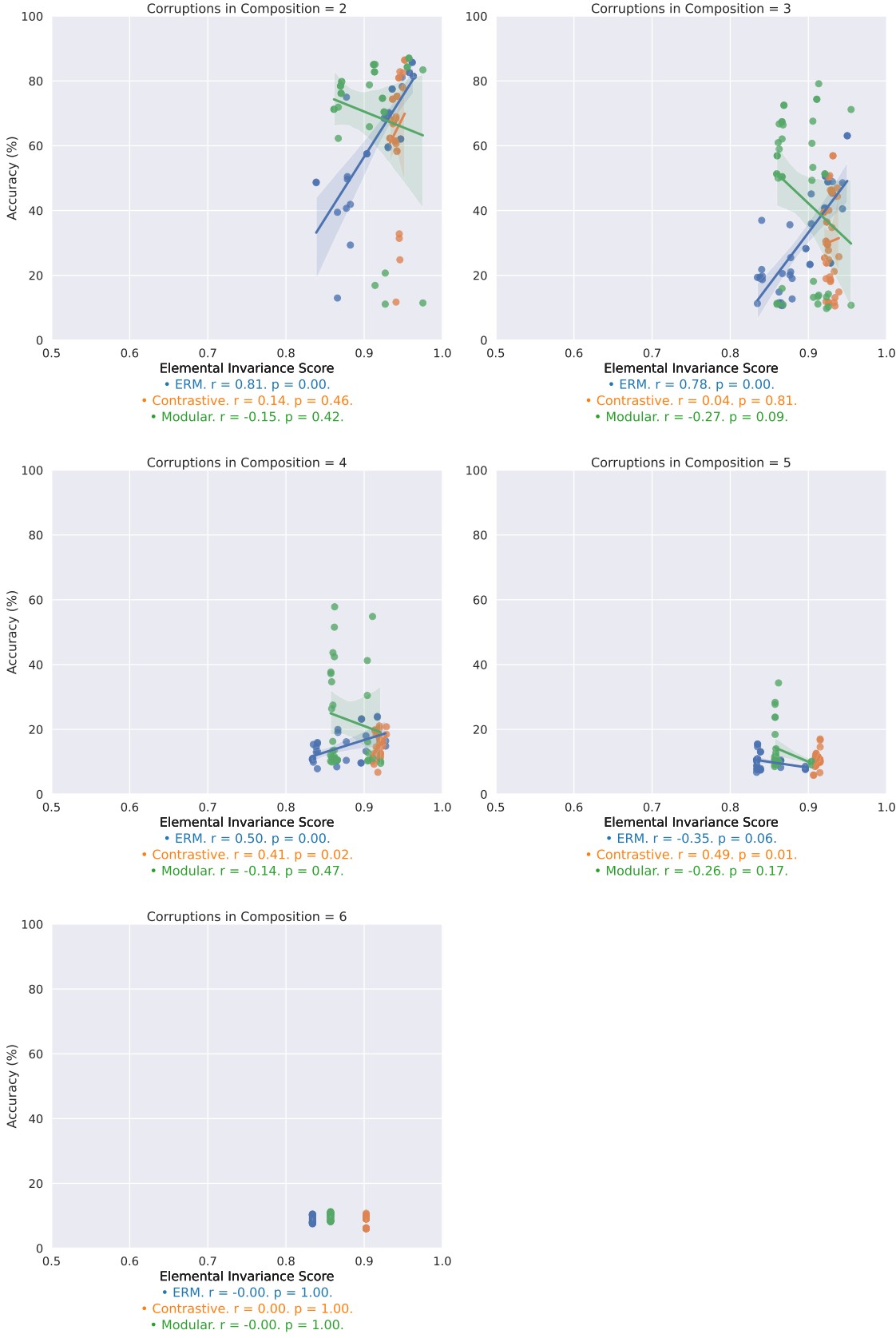

Figure 15: Correlating the elemental invariance score with compositional robustness for CIFAR-10. These plots expand the first row of Figure 10 to show all compositional test domains.

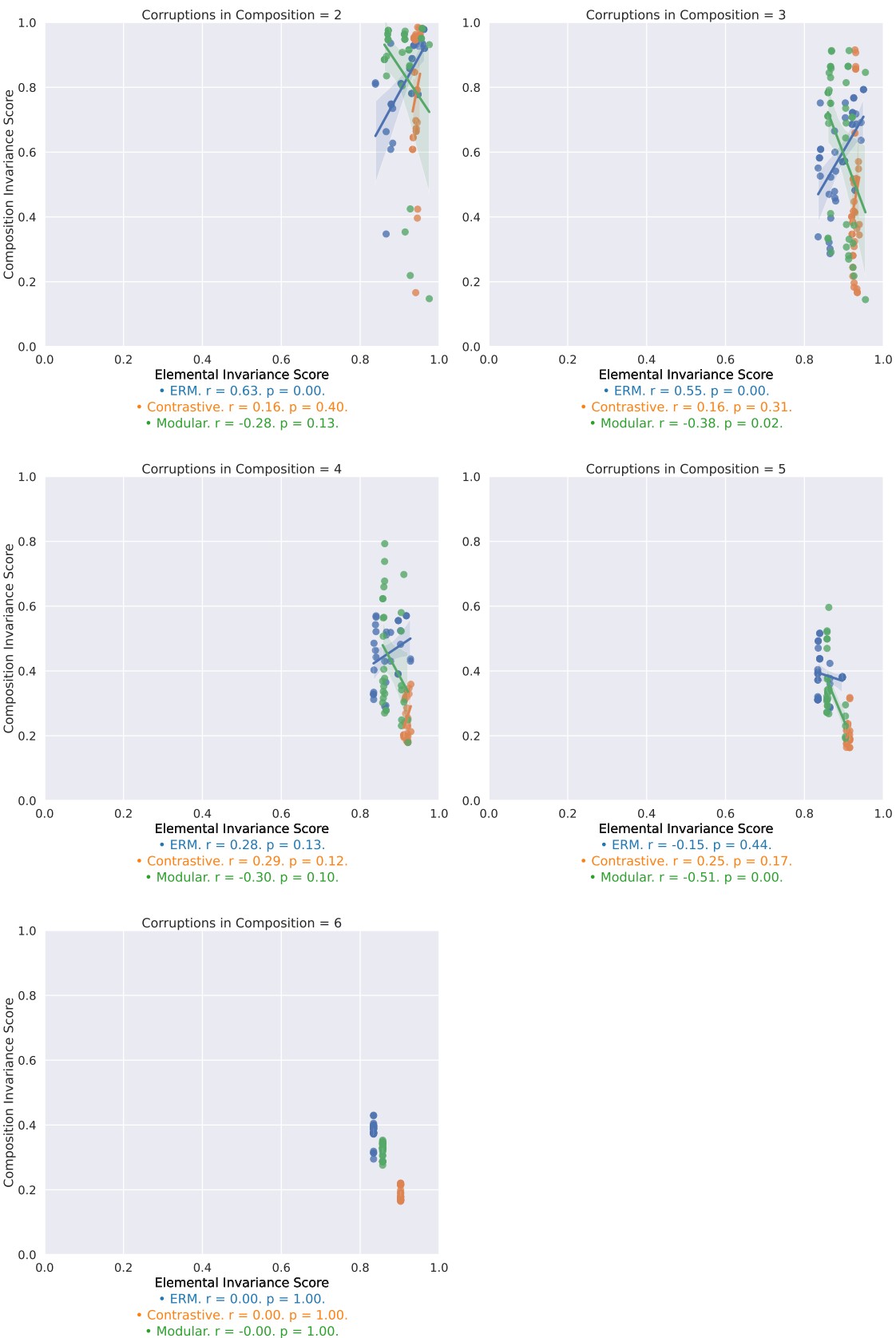

Figure 16: Correlating the elemental invariance score with the composition invariance score for CIFAR-10. These plots expand the second row of Figure 10 to show all compositional test domains.

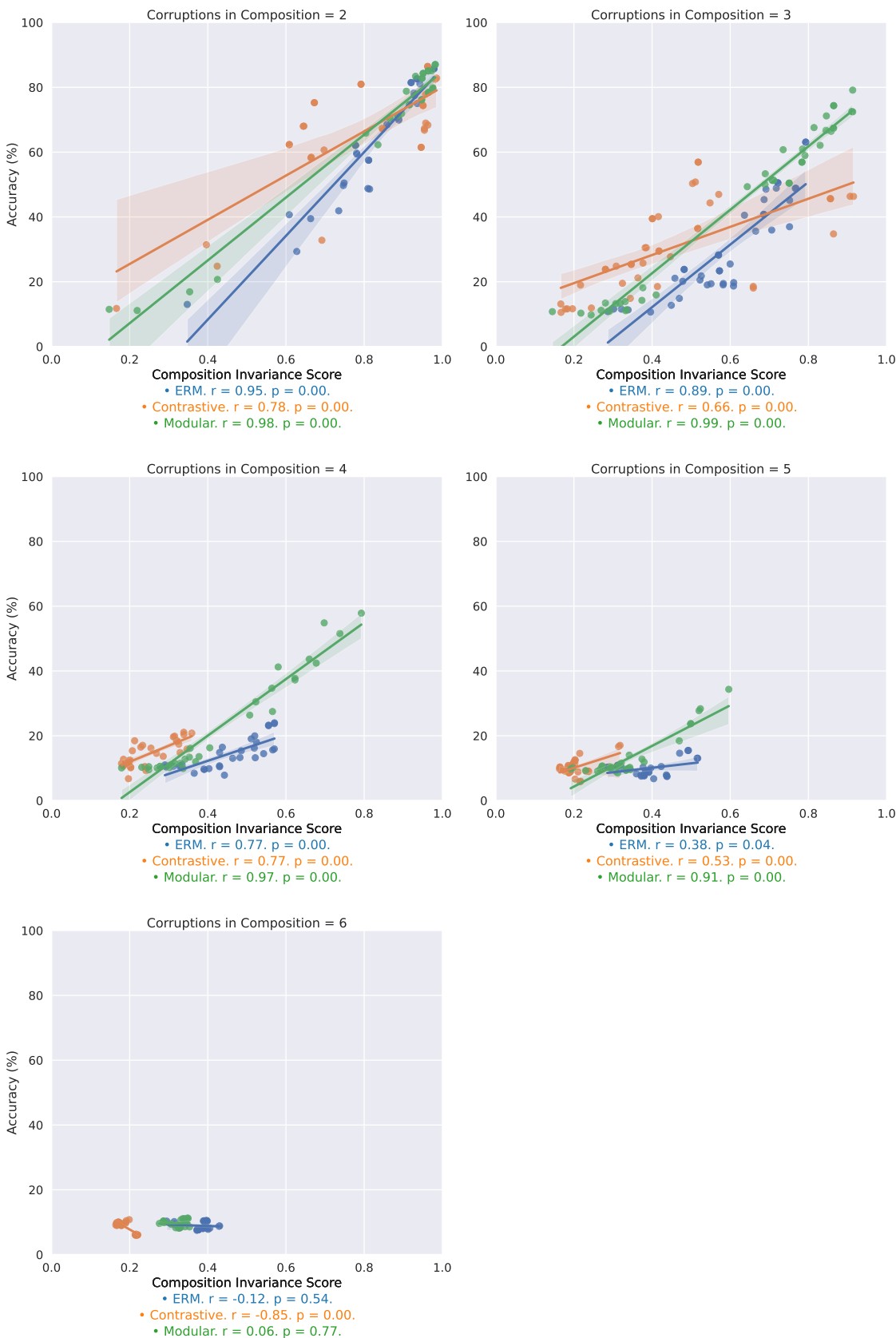

Figure 17: Correlating the composition invariance score with compositional robustness for CIFAR-10. These plots expand the third row of Figure 10 to show all compositional test domains.

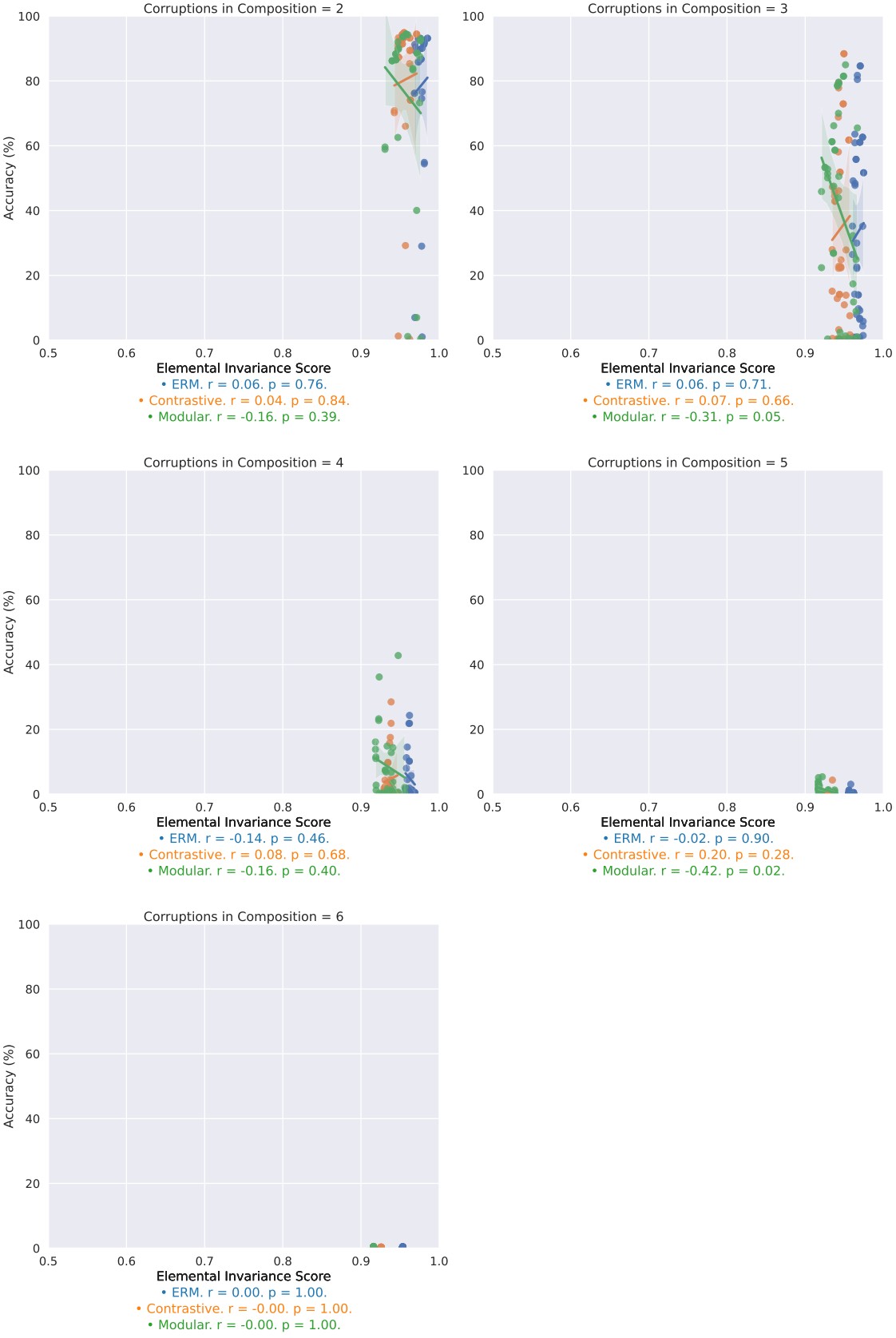

Figure 18: Correlating the elemental invariance score with compositional robustness for FACESCRUB. These plots expand the first row of Figure 11 to show all compositional test domains.

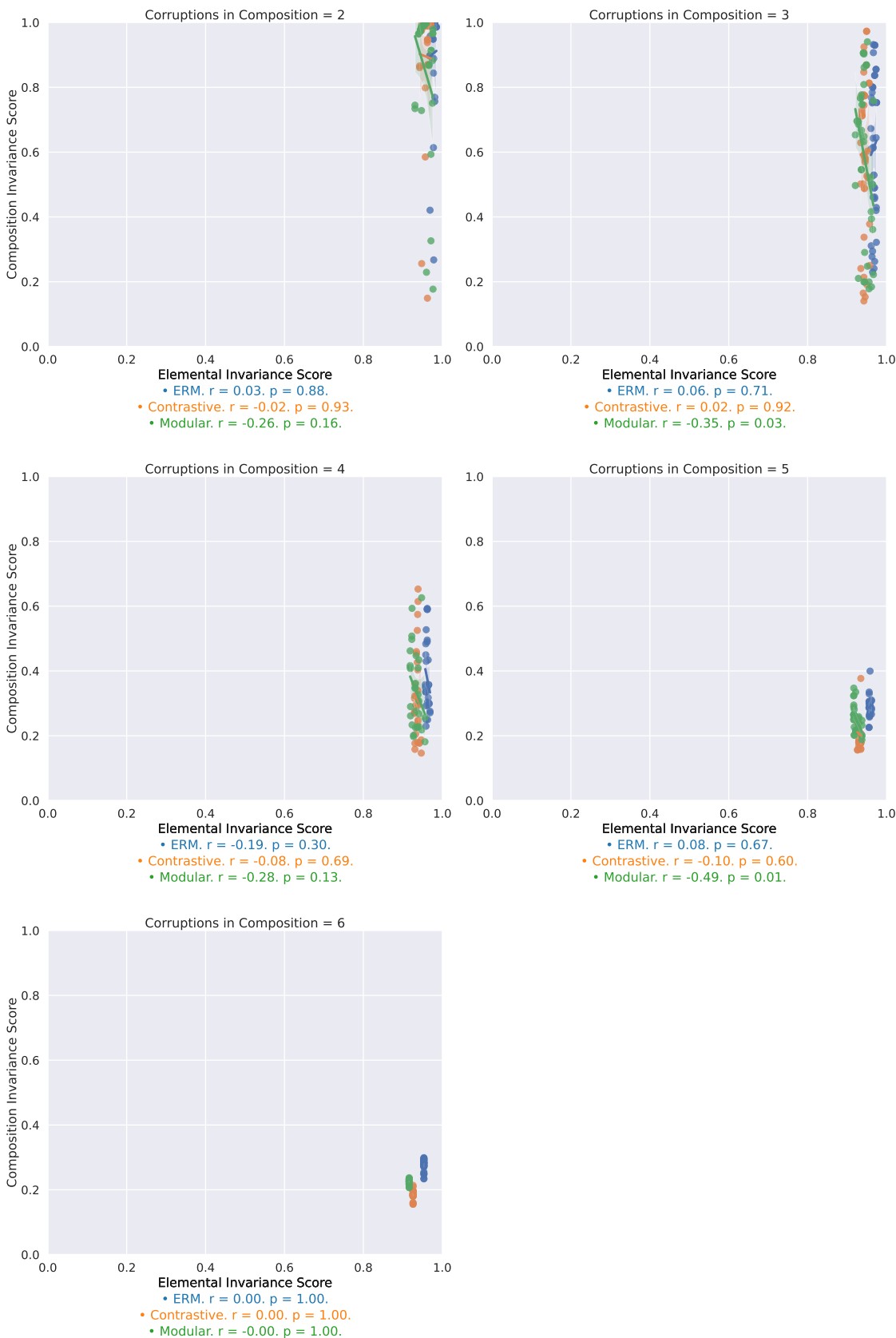

Figure 19: Correlating the elemental invariance score with the composition invariance score for FACESCRUB. These plots expand the second row of Figure 11 to show all compositional test domains.

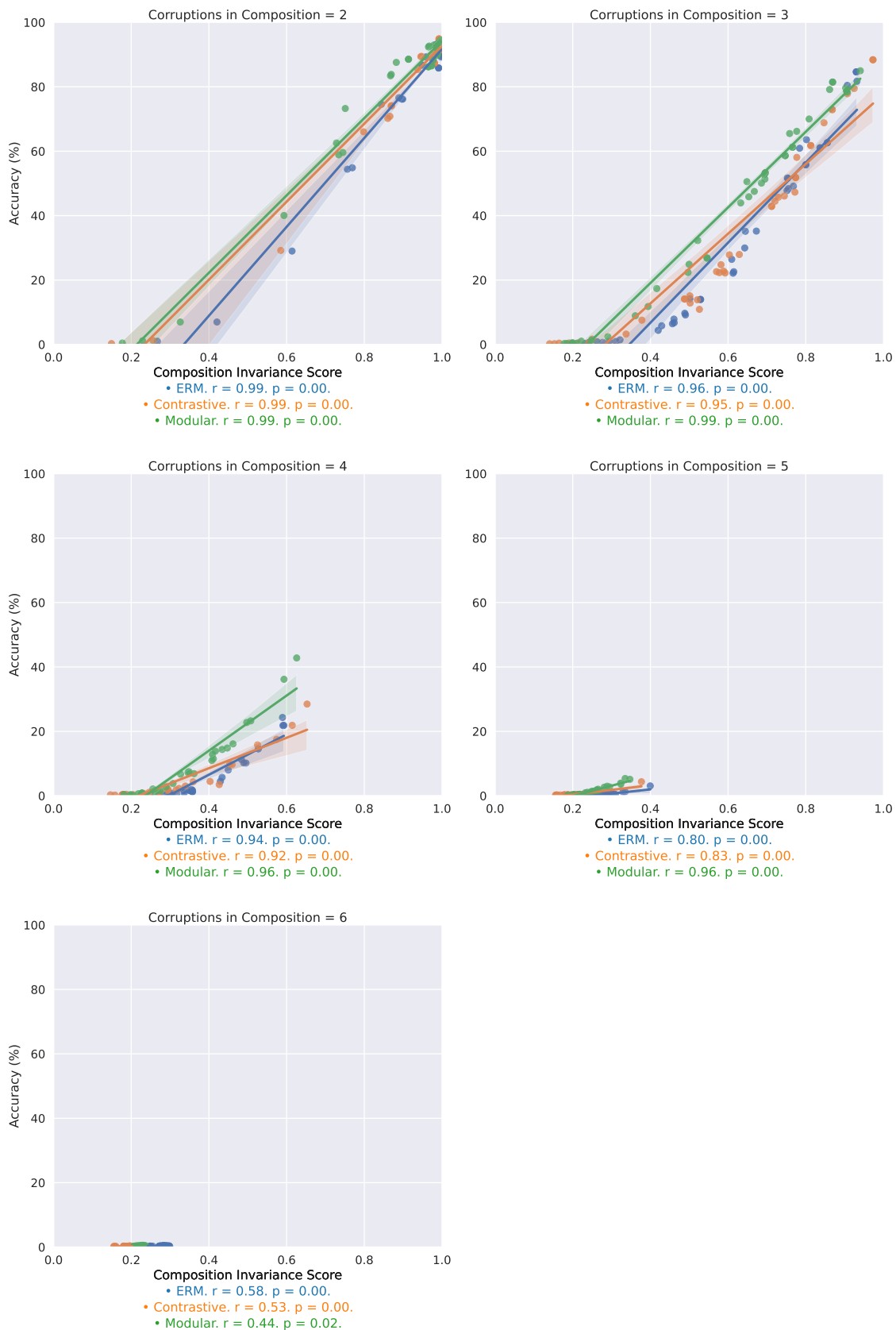

Figure 20: Correlating the composition invariance score with compositional robustness for FACESCRUB. These plots expand the third row of Figure 11 to show all compositional test domains.

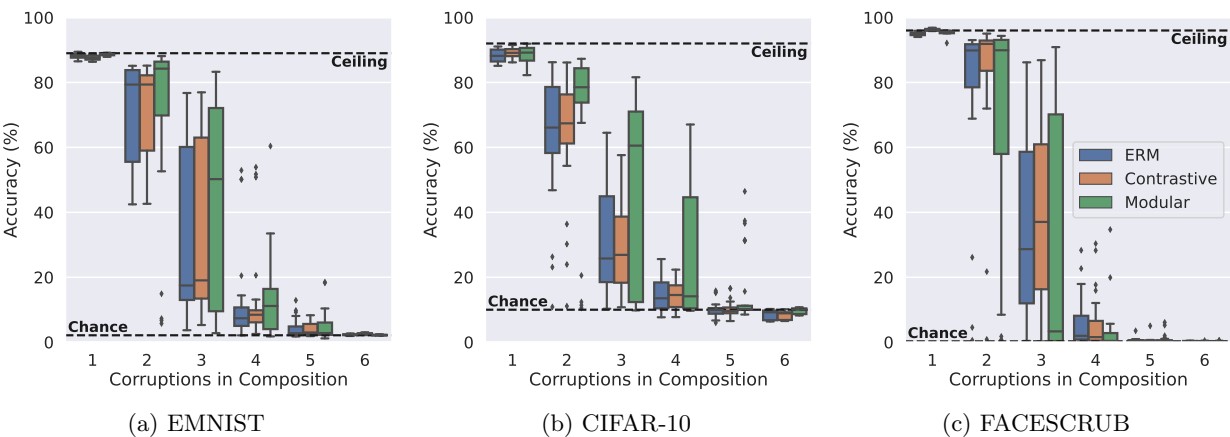

(a) EMNIST            (b) CIFAR-10            (c) FACESCRUB

Figure 21: Evaluating compositional robustness on different datasets (second random seed). This figure is the same as Figure 2 with a different random seeding.

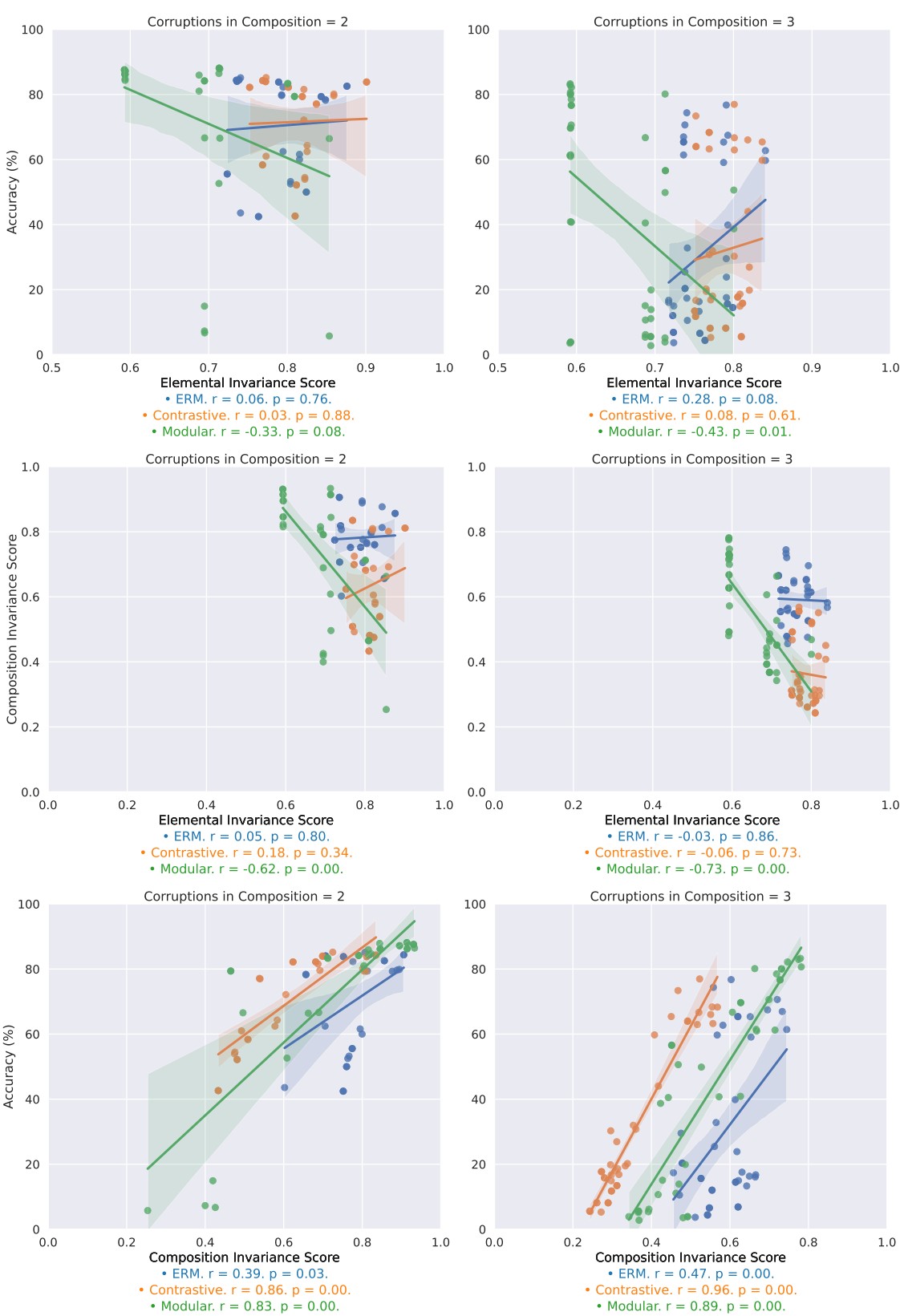

Figure 22: Correlating invariance scores with compositional robustness for EMNIST (second random seed). This is the same as Figure 3 with a different random seeding.

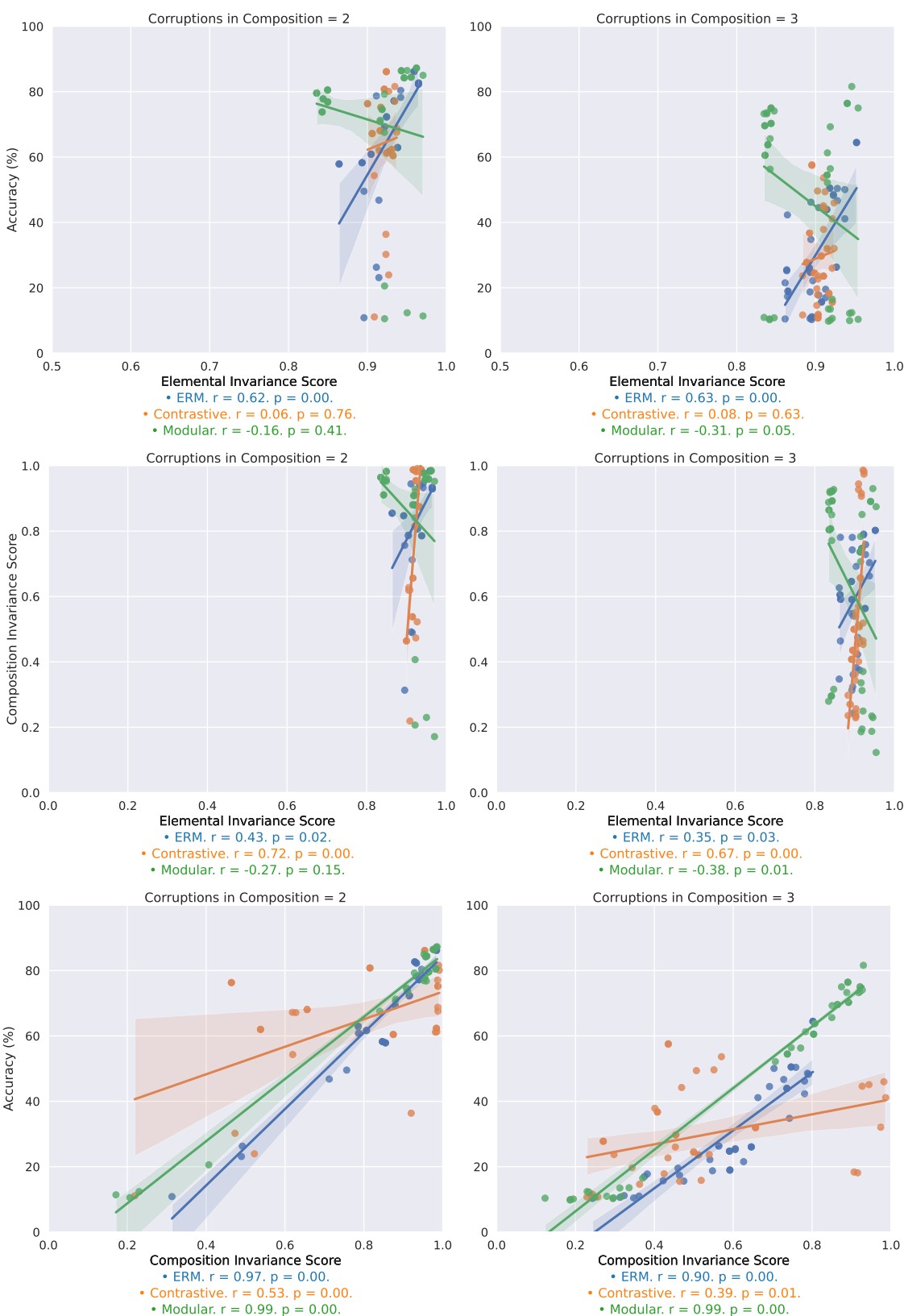

Figure 23: Correlating invariance scores with compositional robustness for CIFAR-10 (second random seed). This is the same as Figure 10 with a different random seeding.

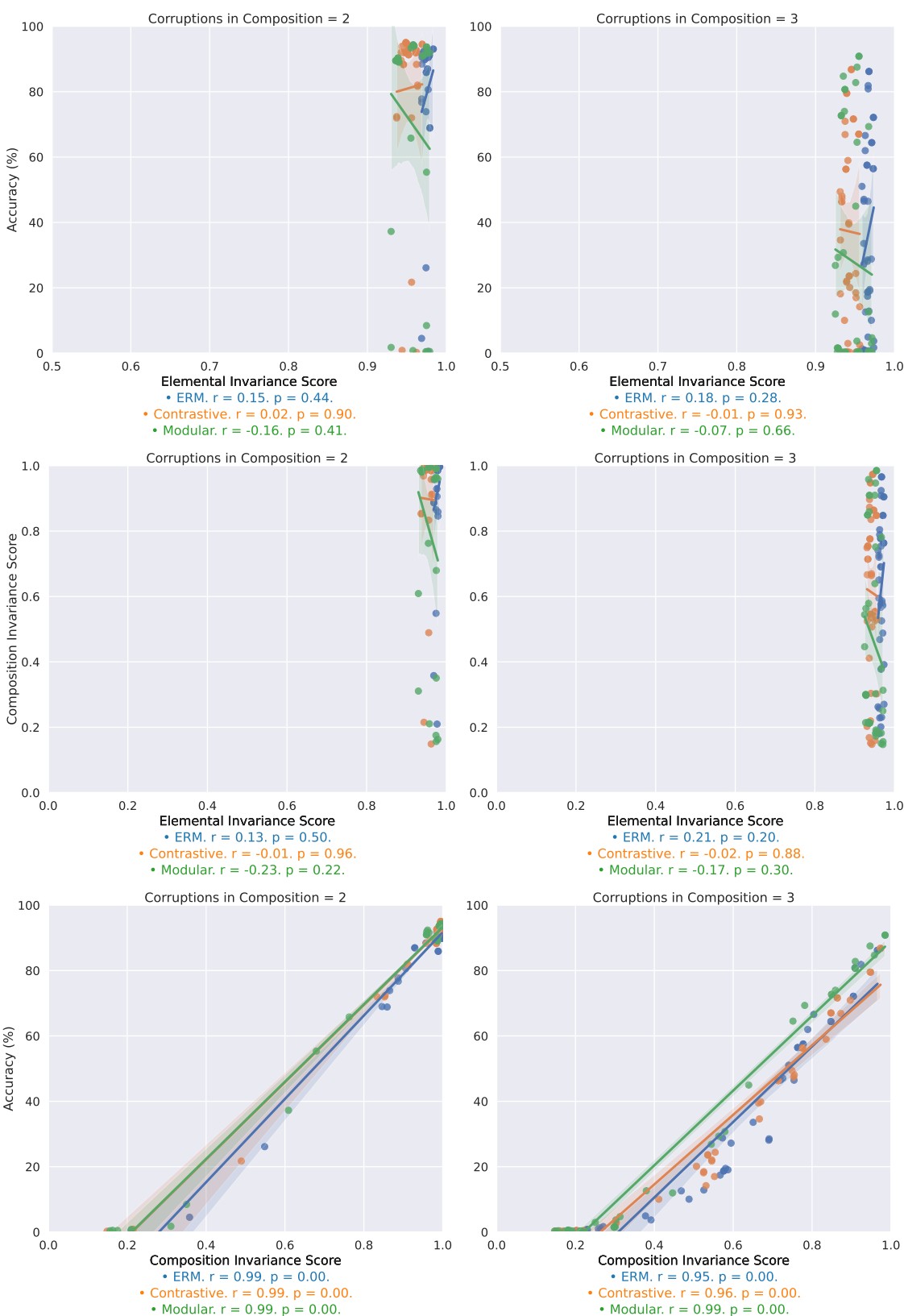

Figure 24: Correlating invariance scores with compositional robustness for FACESCRUB (second random seed). This is the same as Figure 11 with a different random seeding.

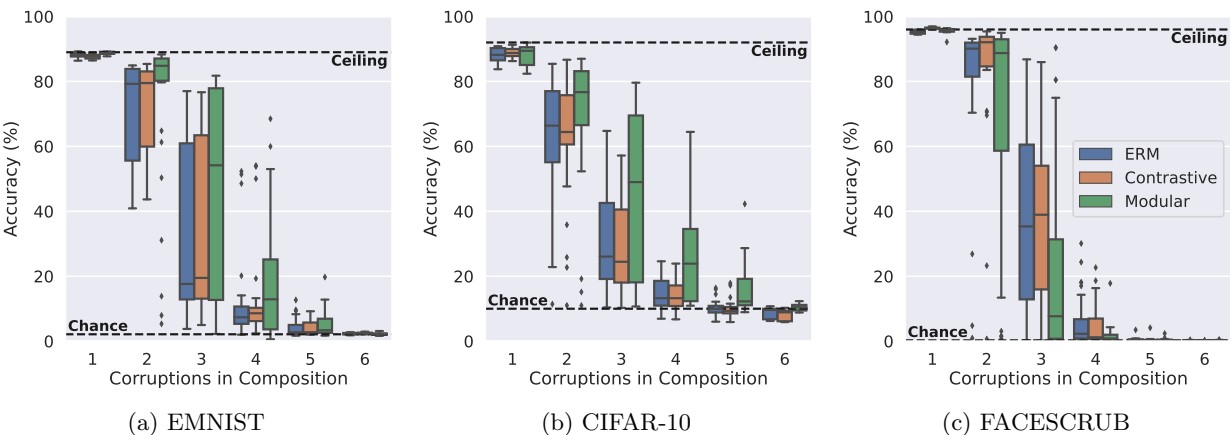

(a) EMNIST                    (b) CIFAR-10                    (c) FACESCRUB

Figure 25: Evaluating compositional robustness on different datasets (third random seed). This figure is the same as Figure 2 with a different random seeding.

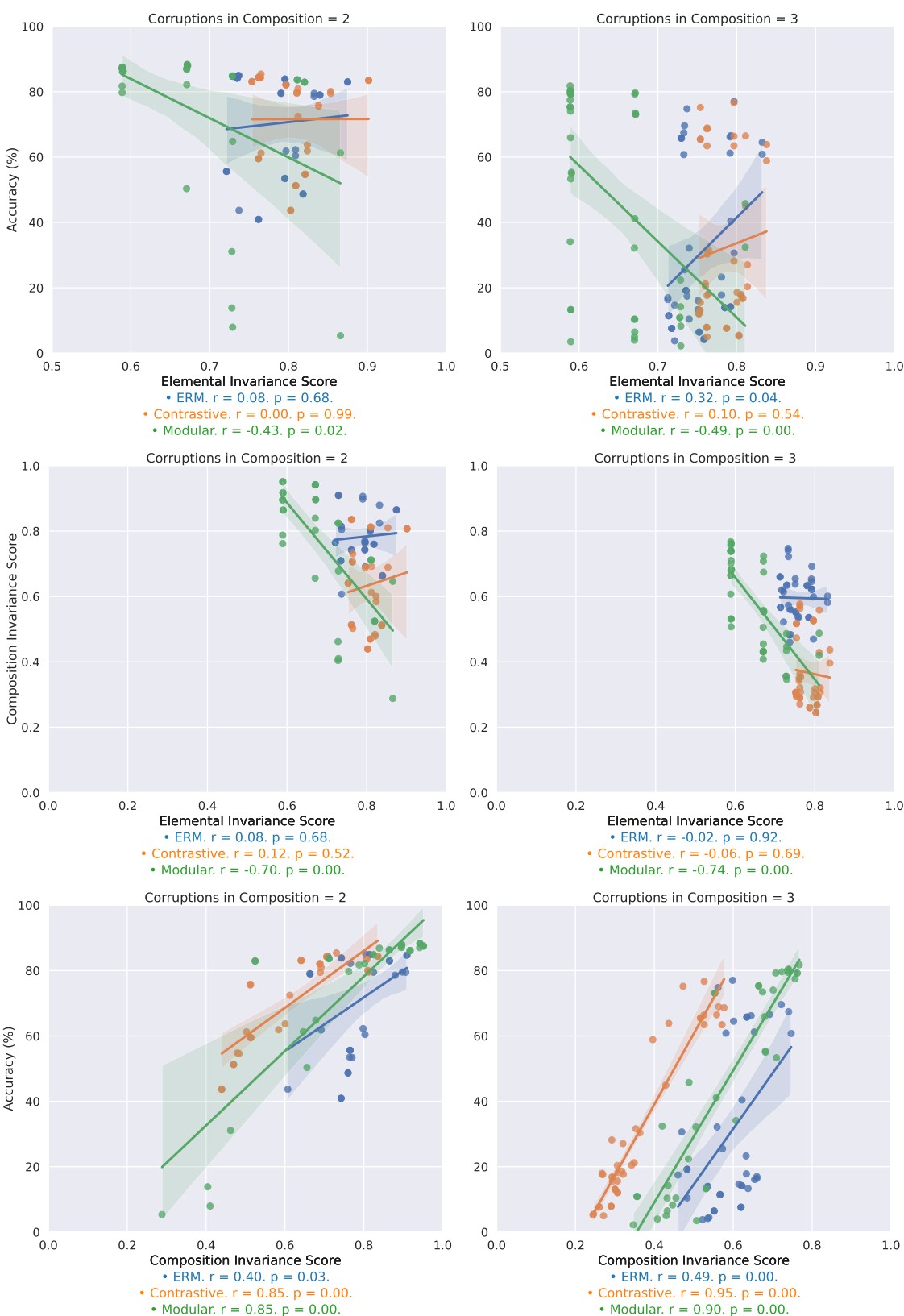

Figure 26: Correlating invariance scores with compositional robustness for EMNIST (third random seed). This is the same as Figure 3 with a different random seeding.

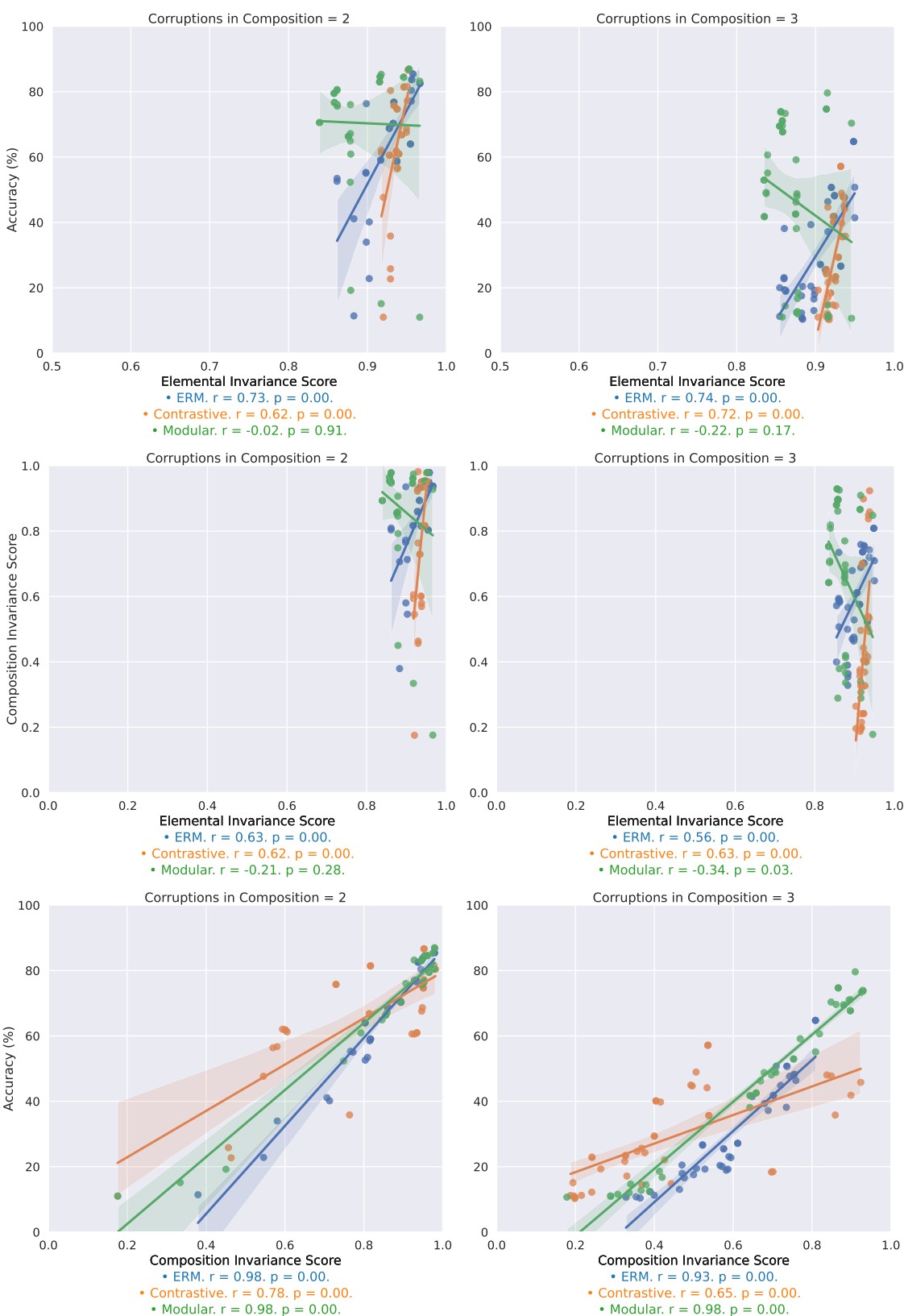

Figure 27: Correlating invariance scores with compositional robustness for CIFAR-10 (second random seed). This is the same as Figure 10 with a different random seeding.

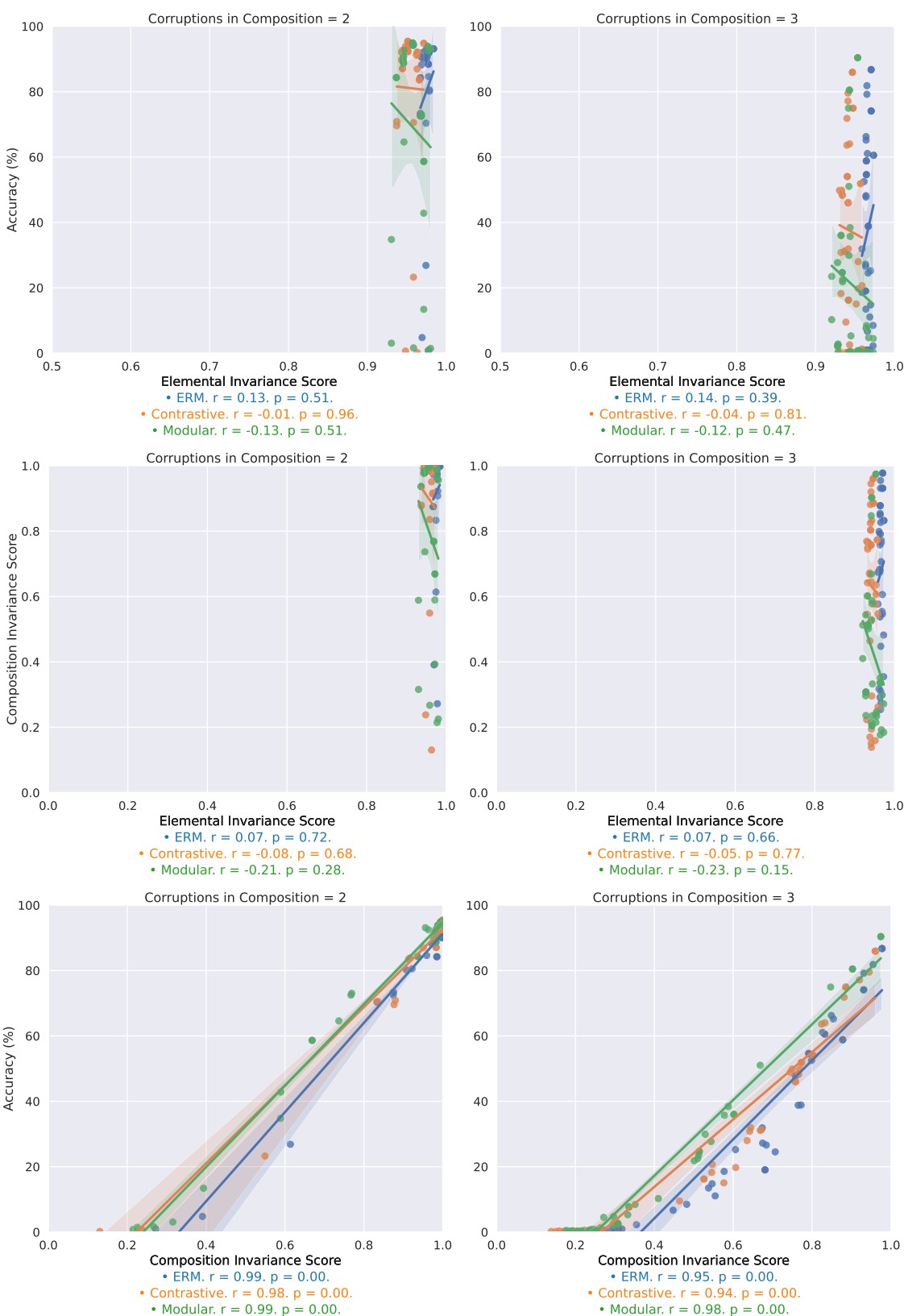

Figure 28: Correlating invariance scores with compositional robustness for FACESCRUB (second random seed). This is the same as Figure 11 with a different random seeding.

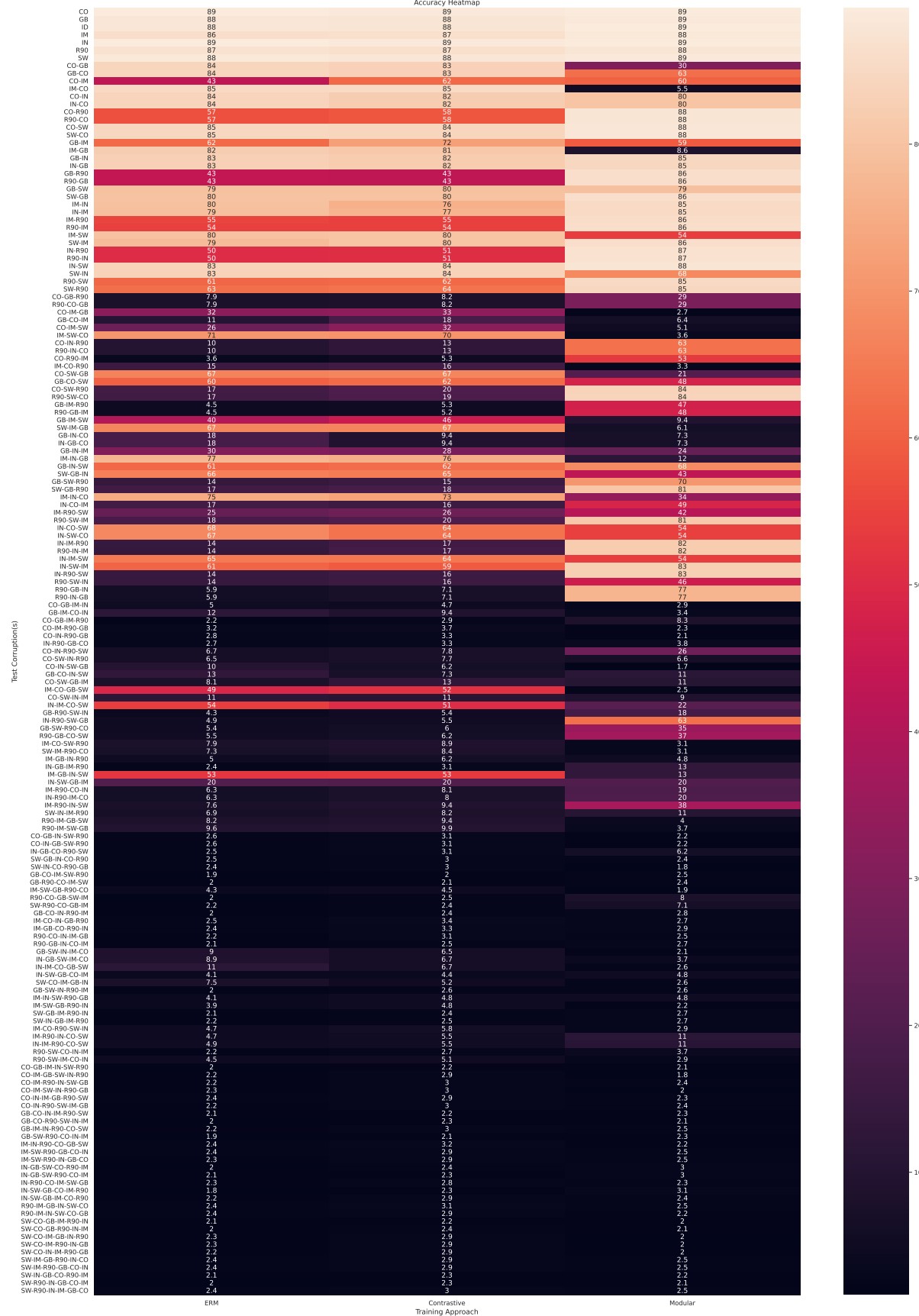

Figure 29: Per-domain heat map for EMNIST (first random seed). This shows the raw EMNIST data from Figure 2. Best viewed with zoom.

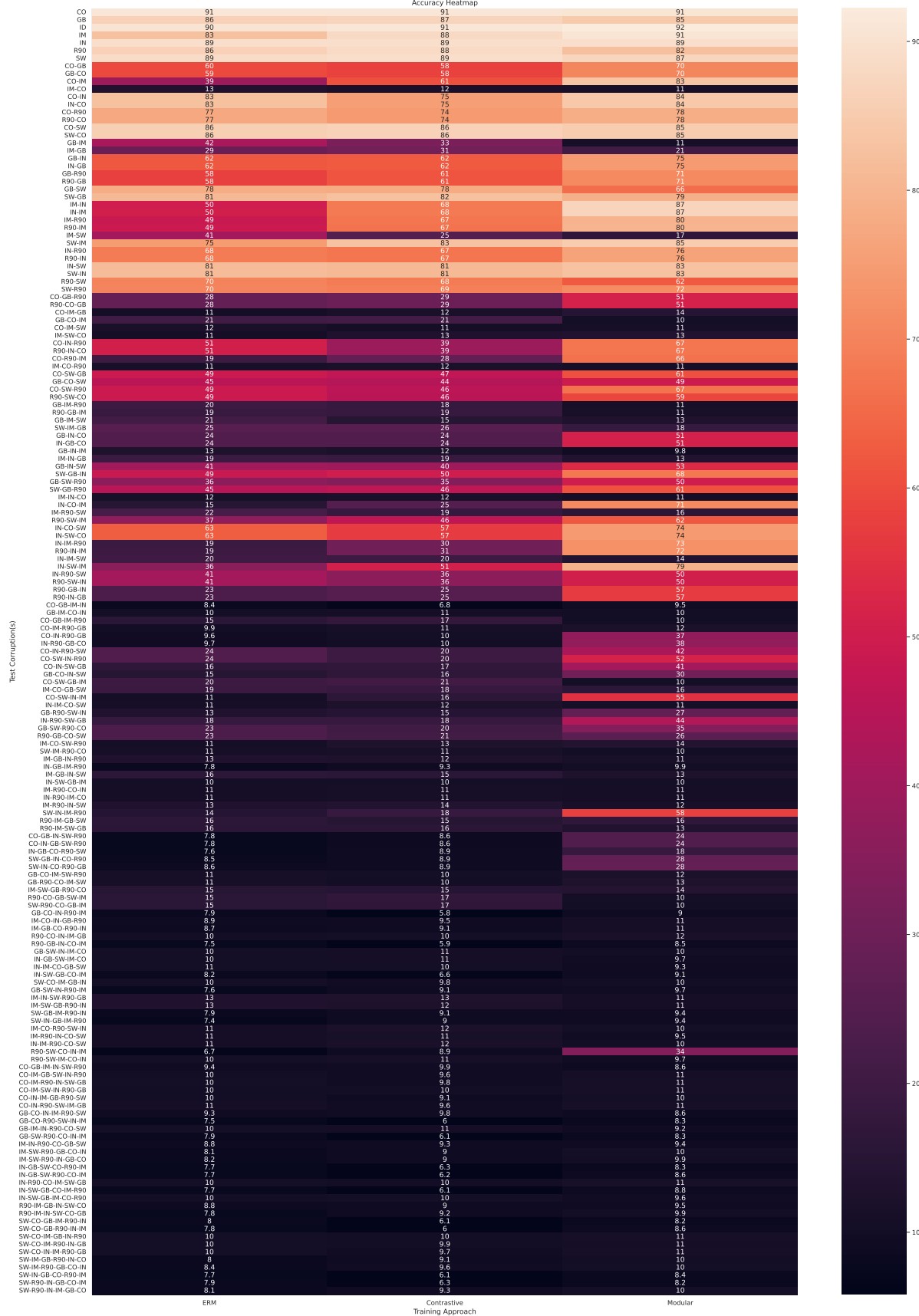

Figure 30: Per-domain heat map for CIFAR-10 (first random seed). This shows the raw CIFAR-10 data from Figure 2. Best viewed with zoom.

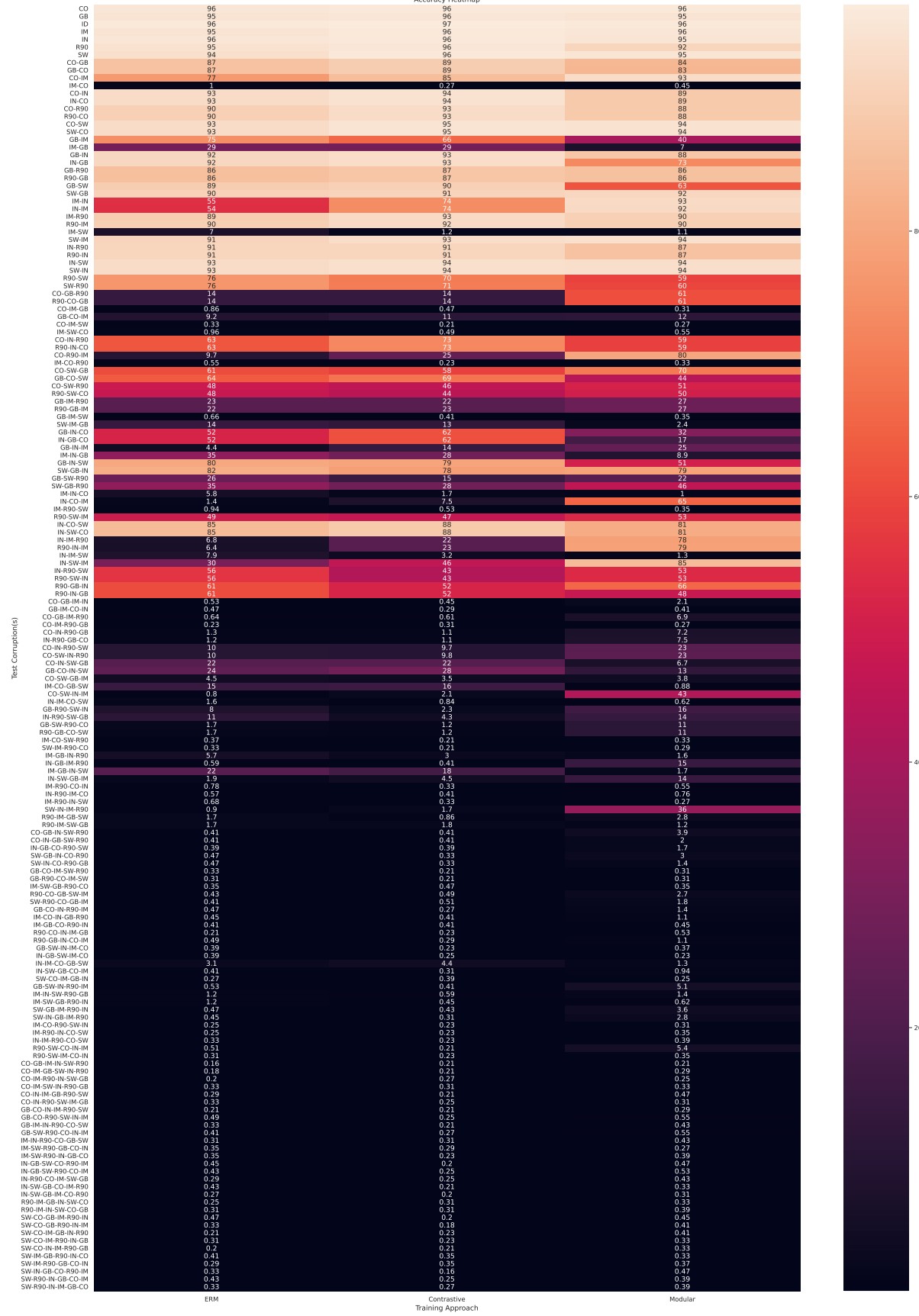

Figure 31: Per-domain heat map for FACESCRUB (first random seed). This shows the raw FACESCRUB data from Figure 2. Best viewed with zoom.

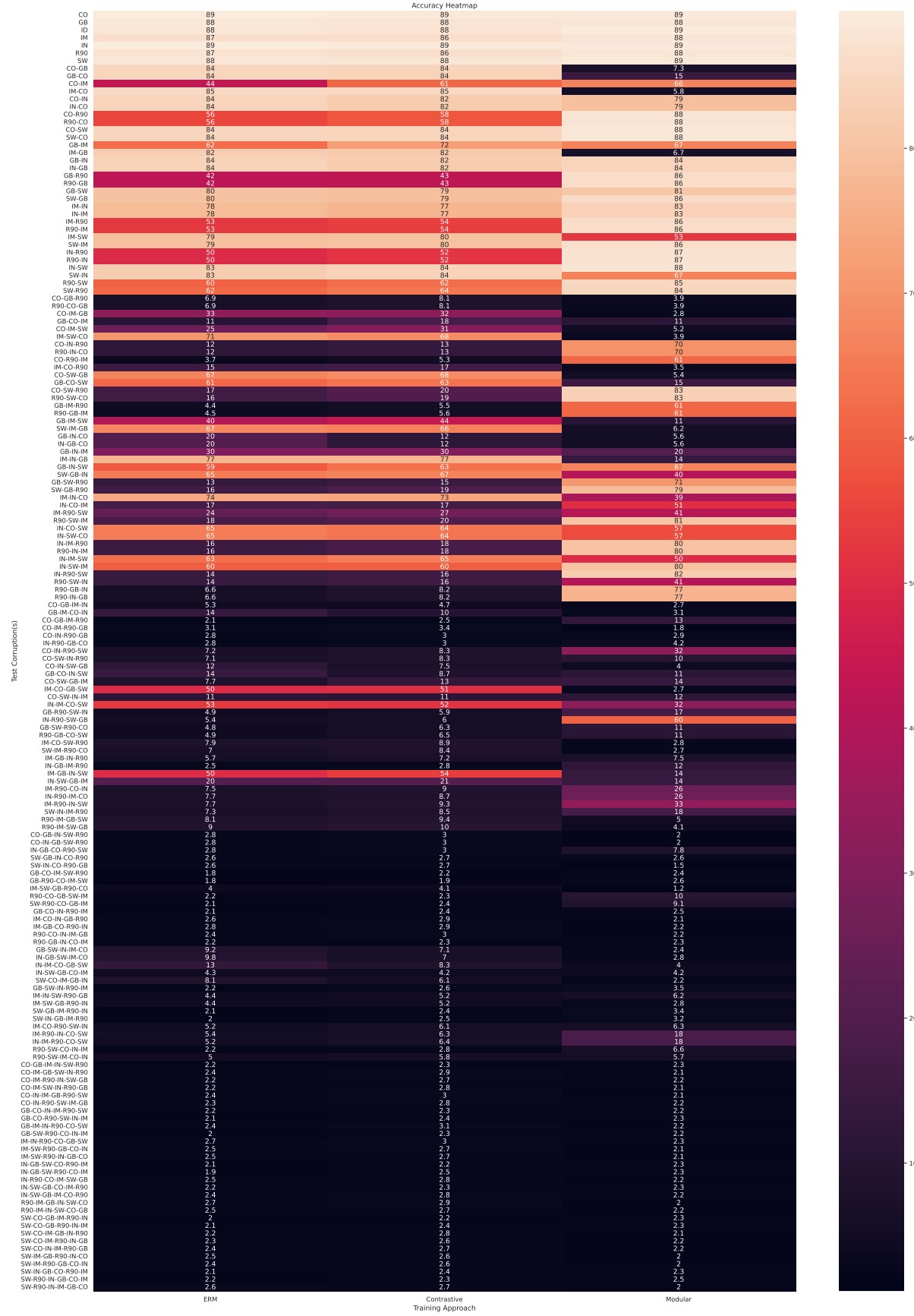

Figure 32: Per-domain heat map for EMNIST (second random seed). This shows the raw EMNIST data from Figure 21. Best viewed with zoom.

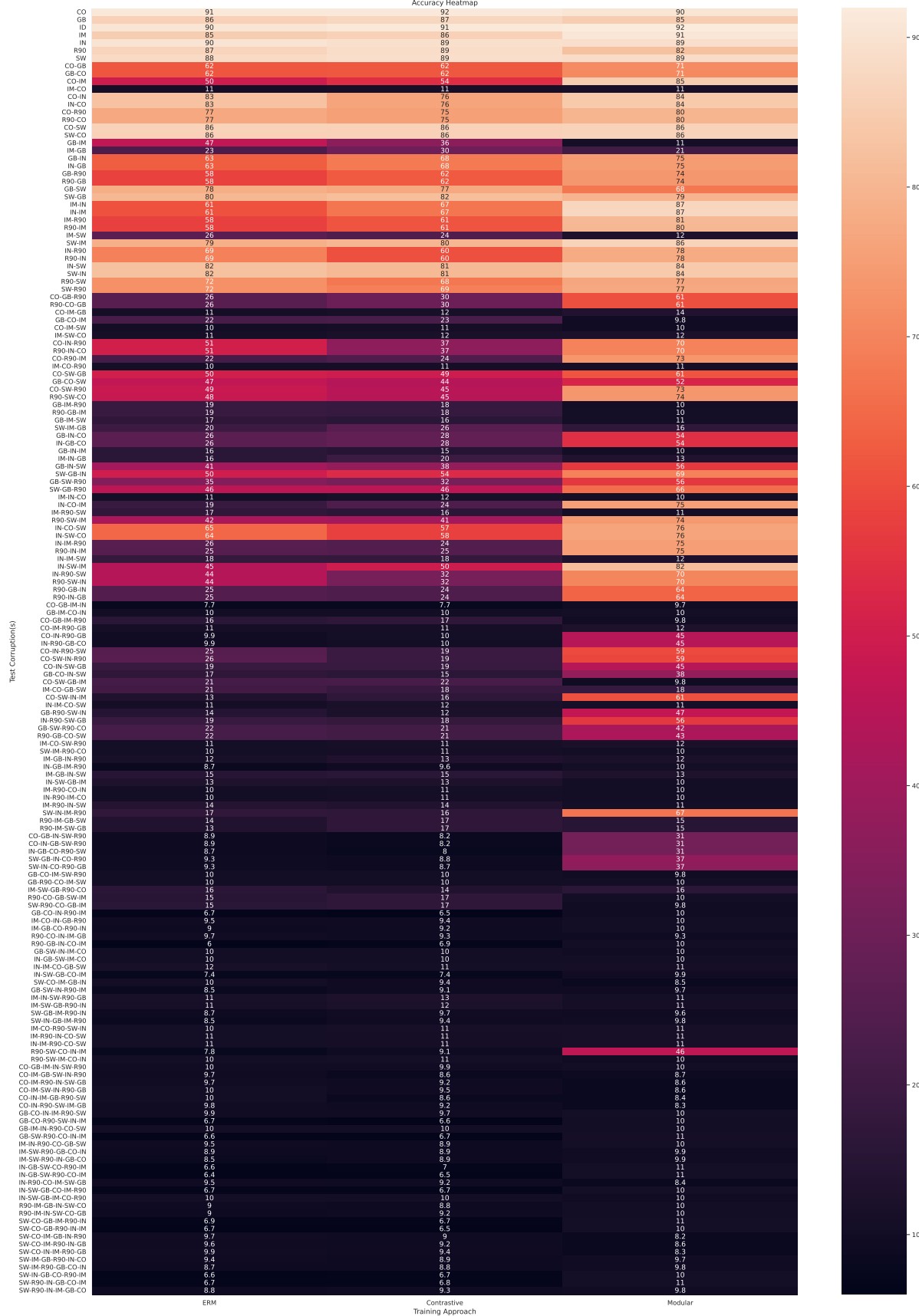

Figure 33: Per-domain heat map for CIFAR-10 (second random seed). This shows the raw CIFAR-10 data from Figure 21. Best viewed with zoom.

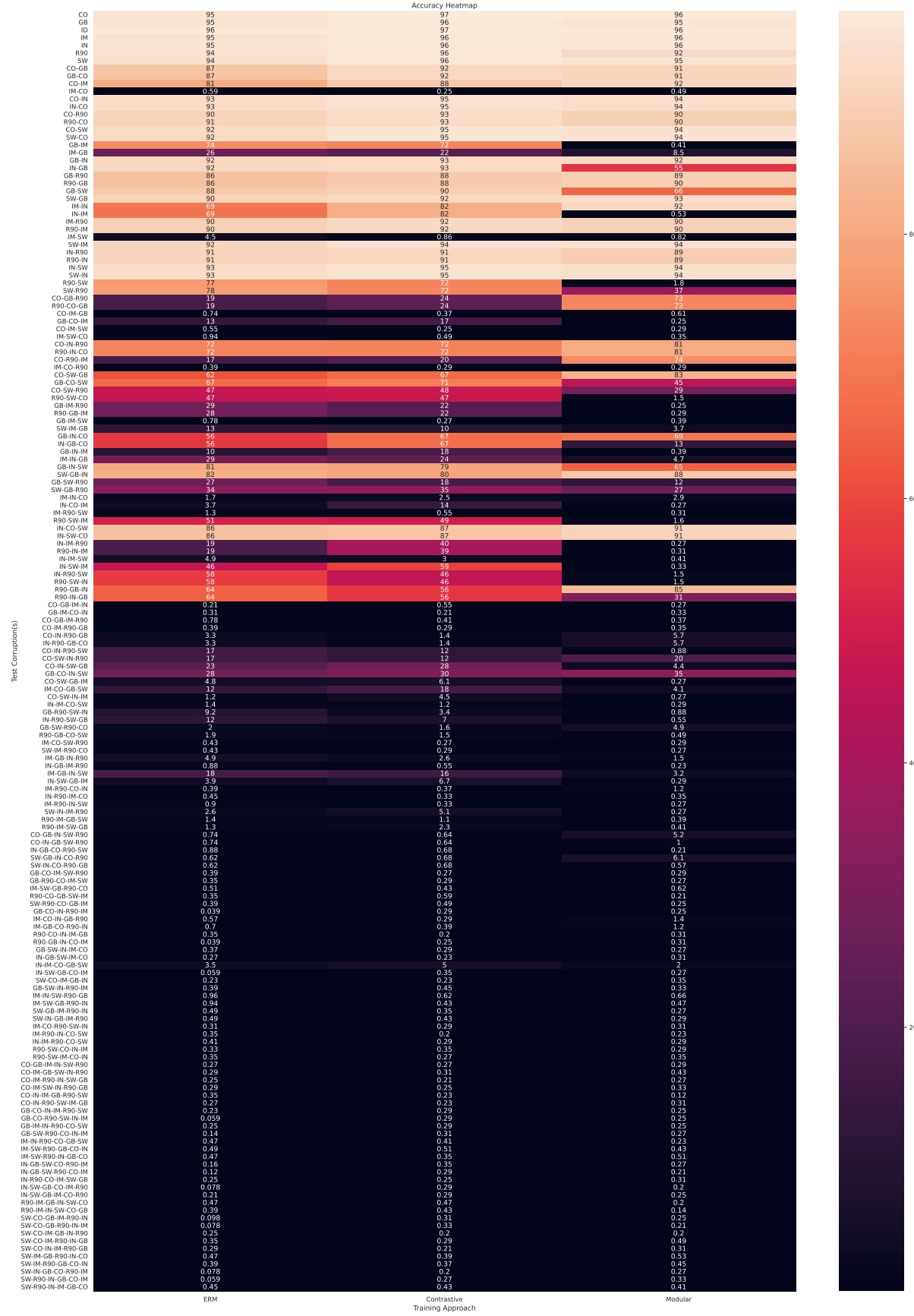

Figure 34: Per-domain heat map for FACESCRUB (second random seed). This shows the raw FACESCRUB data from Figure 21. Best viewed with zoom.

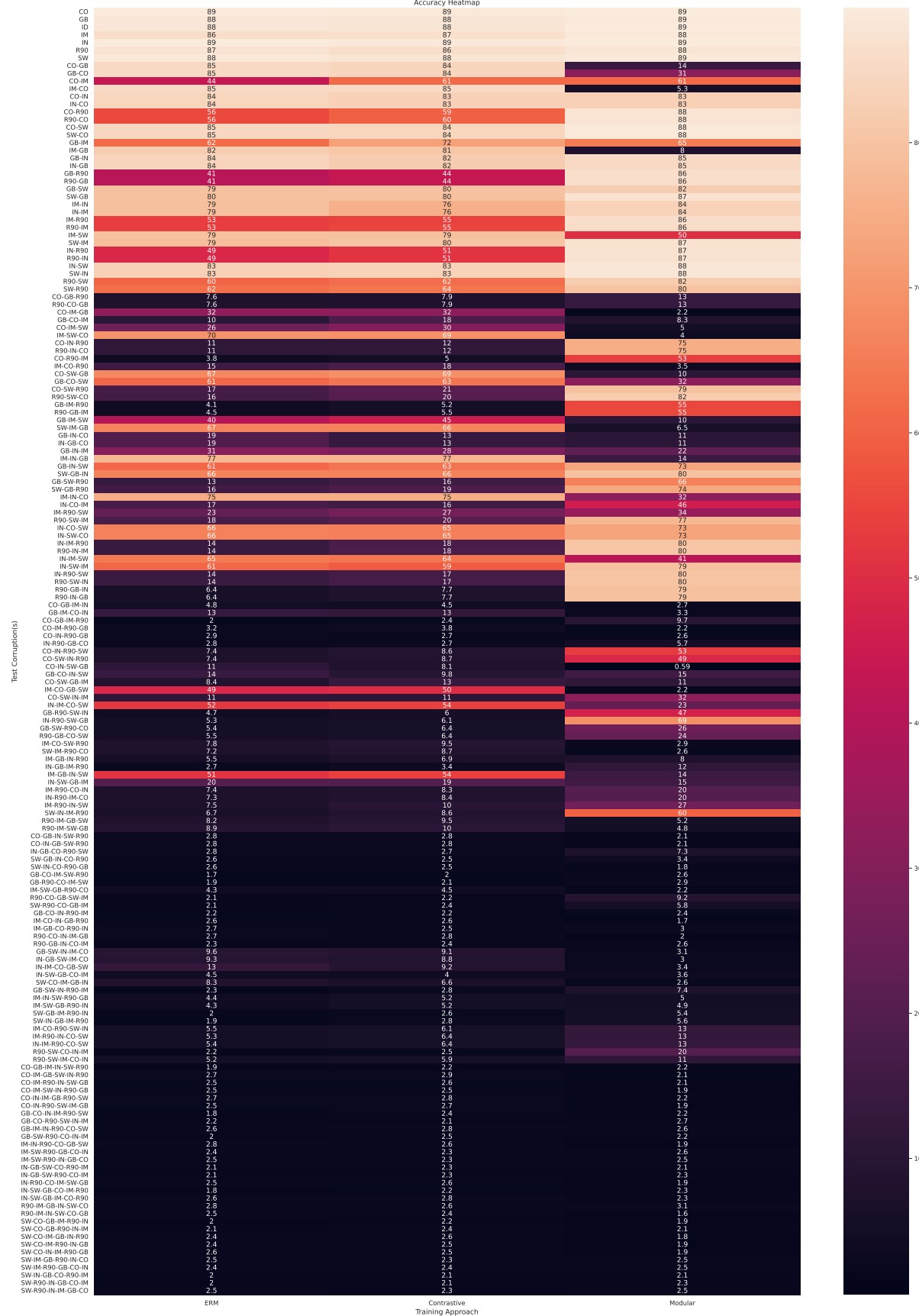

Figure 35: Per-domain heat map for EMNIST (third random seed). This shows the raw EMNIST data from Figure 25. Best viewed with zoom.

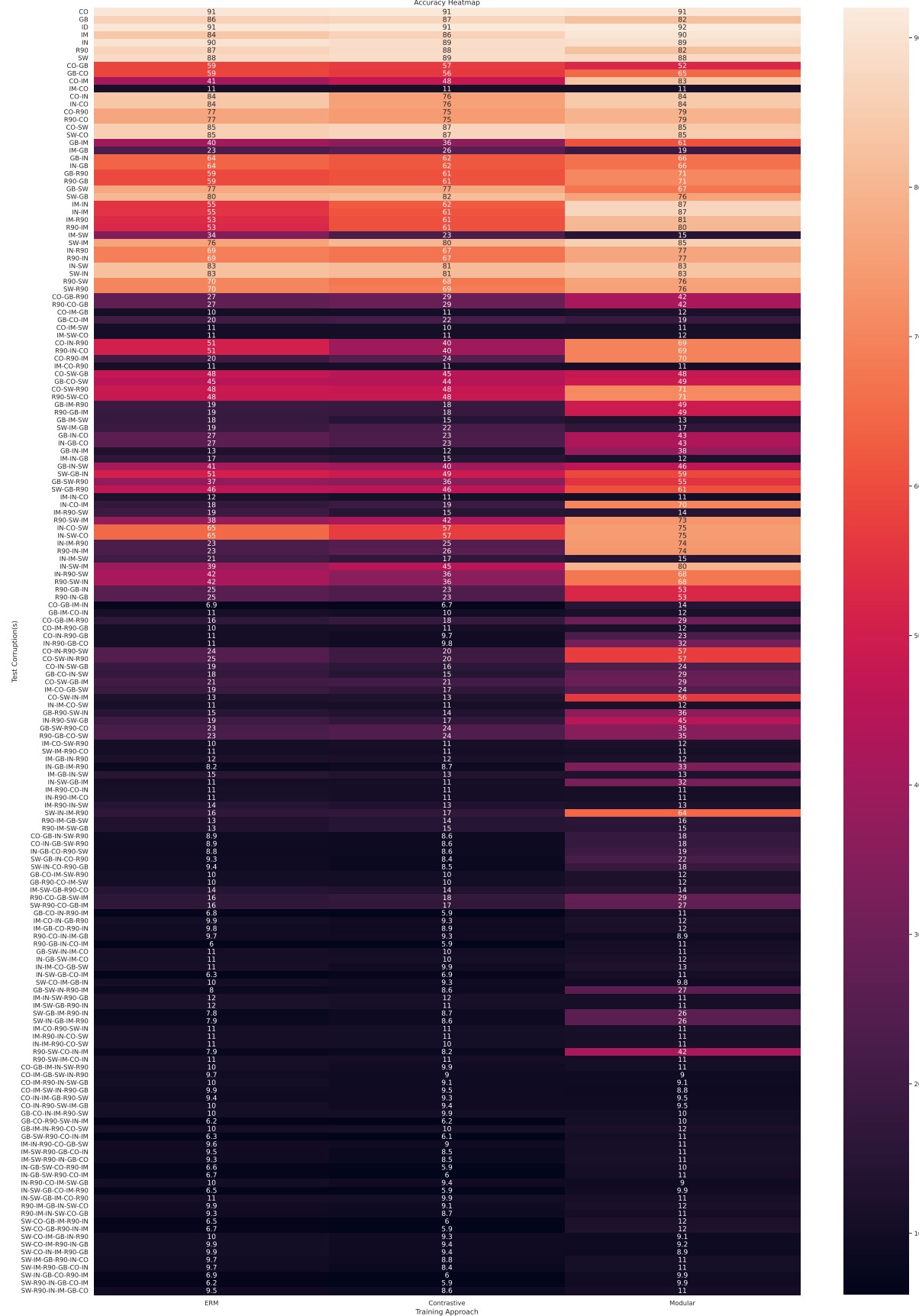

Figure 36: Per-domain heat map for CIFAR-10 (third random seed). This shows the raw CIFAR-10 data from Figure 25. Best viewed with zoom.

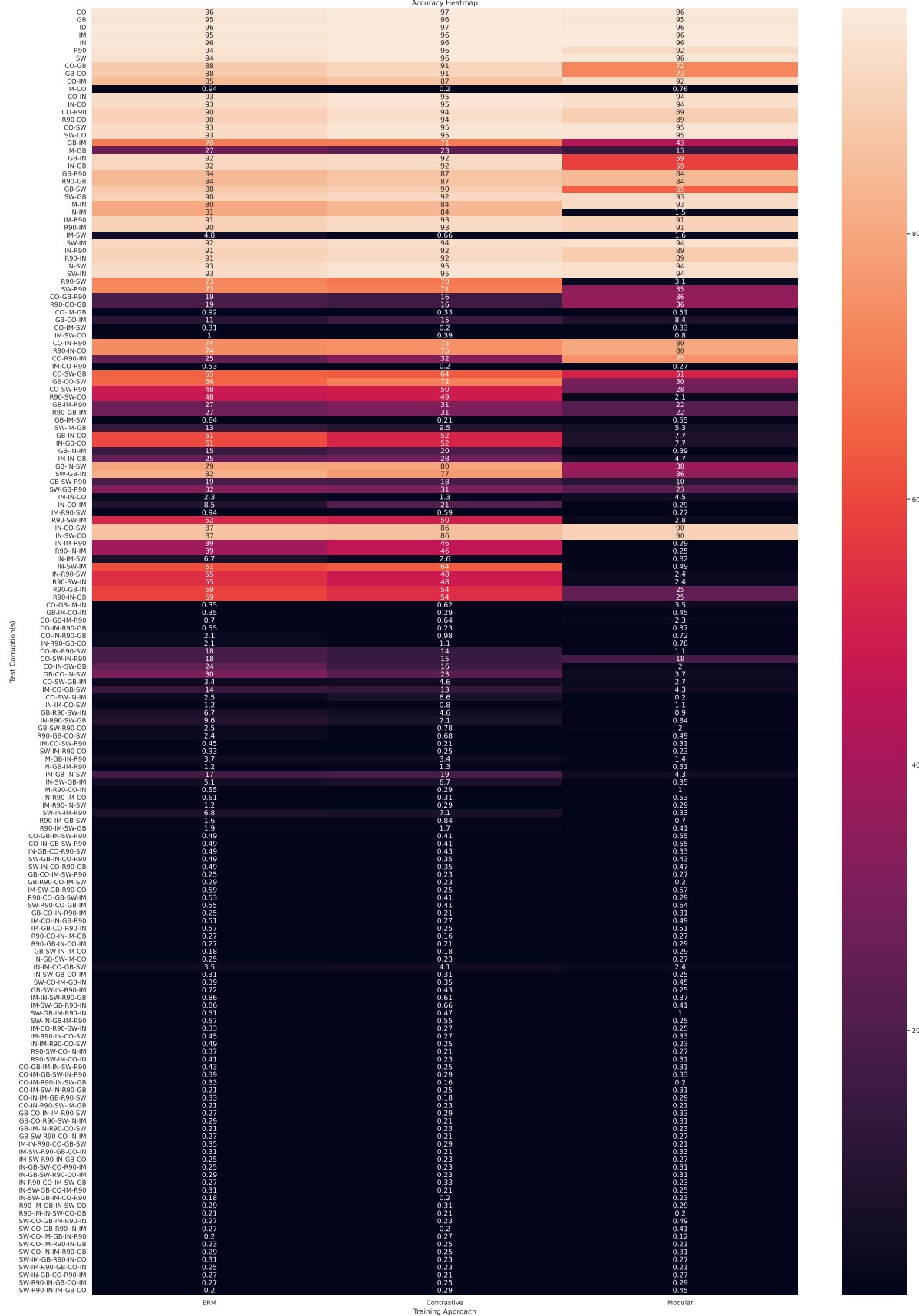

Figure 37: Per-domain heat map for FACESCRUB (third random seed). This shows the raw FACESCRUB data from Figure 25. Best viewed with zoom.

