# OpenReview forum: "Investigations on Modularity and Invariance for Compositional Robustness"
_TMLR — Rejected by TMLR_

### Review · Reviewer_XyV9 · 2024-01-26

**Summary Of Contributions:**

The paper studies compositional robustness to image corruptions, that is: to which extent do methods that were trained on single corruptions generalize to images containing several (2-6) corruptions combined. The authors compare both monolithic deep networks (trained with ERM or with a contrastive loss encouraging invariances) and a modular approach in which layer "experts" specialize to get invariant to certain corruptions. The empirical findings suggest that training for invariances does not substantially help when confronted with composed corruptions, while a modular approach does improve performance (albeit it seem this comparison is not fair, see below).

**Audience:**

Yes

**Broader Impact Concerns:**

no concerns

**Claims And Evidence:**

No

**Requested Changes:**

* Please clarify which severities the single elemental corruptions have (regarding the ImageNetC severity scale).
* Please conduct experiments with lower severity of the elemental corruptions, such that combinations of 4-6 corruptions still allow better than chance-level performance
* Please include a stronger monolithic baseline that utilizes knowledge about the corruptions present in a specific input
* experiments on larger-scale data sets such as ImageNet would strengthen the paper

**Strengths And Weaknesses:**

Strengths:
  * The problem of compositional robustness is well motivated and relevant and should be of interest to TMLR's audience
  * The related work is thoroughly reviewed and discussed  in detail in Section 2
  * The paper is well written and easy to follow
  * Results are nicely presented (Figure 2 and Figure 3) and discussed in depth.
  * Limitations of the work are acknowledged

Weaknesses:
 * The proposed modular approach is based on an unrealistic assumption, namely that an oracle knows which corruptions are present in an image. This information is used to enable/disable experts that were trained for compensating the corresponding elemental corruptions. In my opinion, this is not a useful setting. The authors also acknowledge this limitation at the end of Section 4.4. At the very least, if the experiments are based on assuming that such an oracle exists, the monolithic baselines should also be modified to make use of this knowledge (as discussed in Section 5). Otherwise, one can draw no conclusions from the experiments.
 * The elemental corruptions have high severity such that combinations of 3 or more of them make the image unrecognizable (also to humans, see Figure 1). This is not a reasonable experimental setting since even the strongest method can hardly perform above chance level.
 * The weighting of the two loss terms in Equation 3 is crucial for learning invariant representations. How was $\lambda$ selected?
 * The datasets studied are relatively small (CIFAR10, EMNIST etc.) and provide limited insights about more realistic settings.

Minor:
 * The wording "oven-ready method" (Section 4.4) is not appropriate for a scientific paper

---

> ### Author Response · Authors · 2024-02-25
> **Updates & Clarifications**
>
> Thank you for the detailed feedback, we have made some updates and provided some clarifications.
>
> **Paper updates**
>
> We have modified appendix A, adding A.2, to include information on the corruption settings and an additional experiment for EMNIST using the lowest severities. Here we see limited degradation in performance with compositions of 2 corruptions but above chance accuracy with compositions of 6. The relative performance of the different methods is unchanged. The original settings of severity were intentionally chosen to get a significant reduction in performance with compositions of just 2 corruptions, in order to make experimentation and analysis faster and easier to understand.
>
> We have removed "oven-ready" and replaced with "directly applicable".
>
>
> **Clarifications**
>
> The question we are investigating is do models that separately process data generating factors improve over models that do not. We take this question to the extreme in order to perform analysis, making use of an oracle, full modularization of the corruptions and paired data. Based on the TMLR guidelines we think this work falls under "experimental studies yielding new insight into the behavior of learning in intelligent systems" and "formalization of new learning tasks", rather than "new algorithms with sound empirical validation". It is true that the assumption of an oracle in the real world is unrealistic, but that doesn't mean we can learn nothing from the experiments presented.
>
> To elaborate further, the aim of the discussion in section 5 is to highlight that, in this controlled set up, there is a correspondence between modular architectures and knowledge of which corruptions (or mechanisms) are present. A neural network that has as input any constant representation of a corruption can be rewritten as a network containing specific parameters for processing that corruption (a one-hot label is a special case of this which we discuss in the text). That is, we don't see how to provide the corruption information without creating a modular architecture. This observation provides some insight into why modular architectures may work well in the case of [1], [2] or other modular techniques.
>
> As with other hyper-parameters, lambda is grid searched on in-distribution data (elemental corruptions) following the recommendations of [3].
>
> ---
>
> [1] Jacob Andreas, Marcus Rohrbach, Trevor Darrell, and Dan Klein. Neural module networks. In Proceedings of the IEEE conference on computer vision and pattern recognition, pp. 39–48, 2016.
>
> [2] Anirudh Goyal, Alex Lamb, Jordan Hoffmann, Shagun Sodhani, Sergey Levine, Yoshua Bengio, and Bernhard Schölkopf. Recurrent independent mechanisms. In International Conference on Learning Representations, 2021.
>
> [3] Ishaan Gulrajani and David Lopez-Paz. In search of lost domain generalization. In International Conference on Learning Representations, 2022.

---

### Review · Reviewer_Jocc · 2024-01-28

**Summary Of Contributions:**

This paper develops a compositional image classification task where given a few elemental corruptions, models are asked to generalize to compositions of these corruptions to achieve compositional robustness. It empirically shows that the invariance building pairwise contrastive loss achieves only marginal improvements in compositional robustness compared to empirical risk minimization. It then proposes a modular architecture whose structure replicates the compositional nature of the task.  It shows that this modular approach consistently achieves better compositional robustness than non-modular approaches. It also shows that the degree of invariance between representations of "in-distribution" elemental corruptions fails to correlate with robustness to "out-of-distribution" compositions of corruptions.

**Audience:**

Yes

**Broader Impact Concerns:**

There are no concerns about the ethical implications of the work.

**Claims And Evidence:**

No

**Requested Changes:**

1. Provide justifications for the proposed compositional robustness. In particular, discuss if the classes of the images will change or not after combining different corruptions.

2. Perform experiments on some real-world datasets where compositional corruptions are presented.

3. Include an illustration figure to explain the proposed modular approach in section 3.3. Combine Tables 1, 2, and 3 into one table.

**Strengths And Weaknesses:**

I think this paper has the following strengths:

1. The proposed modular approach seems novel and achieves good compositional robustness.

2. It clearly states the practical limitations of the work and discusses different interpretations of the work.

3. It might motivate future work to study compositional robustness.

However, I think this paper has the following weaknesses:

1. The proposed framework for evaluating compositional robustness lacks justifications. I think it needs to define elemental corruption and construct different elemental corruptions in a principal way. For the compositional corruption, it needs to ensure that the combination of several elemental corruptions won't change the classes of the images. It doesn't discuss if the classes of the images will change or not after combining different corruptions. In Figure 1, it seems compositions of corruptions will change the semantic meaning of the images and thus the classes of the images might change. If the classes of the images change, then enforcing compositional robustness is meaningless.

2. The test datasets with compositional corruptions are constructed artificially. It would be good to add some real-world datasets where compositional corruptions are presented. This can justify whether the proposed method works in real-world scenarios.

3. The writing needs to be improved. In section 3.3, it would be good to include an illustration figure to explain the proposed modular approach. Tables 1, 2, and 3 can be combined into one table.

---

> ### Author Response · Authors · 2024-02-25
> **Updates & Clarifications**
>
> Thank you for taking the time to review, we have made some changes to the writing thanks to your comments and provided some additional clarifications below.
>
> **Paper updates**
>
> Thank you for the writing and formatting suggestions, we have created a combined table in section 4. A figure showing the modular approach along with additional description is shown in Appendix B.
>
> **Clarifications**
>
> As discussed in Section 4.4, the aim of this investigation is not to provide a new method for real world data. Instead we note two major threads for improving robustness in the literature - namely invariance and modularity. These can be seen in both methodological approaches ([1] vs. [3]) and causal theoretical approaches ([2] vs [4]). In this paper we create an experimental set up to investigate the differences between these approaches in a scientific style where we try and control the data as much as possible to empirically explore the behaviors of the different approaches.
>
> It is not clear to us what additional justification is needed for the compositional framework. The elemental corruptions we use are directly inspired by existing robustness datasets Imagenet-C, CIFAR-C & MNIST-C, however, the specific functional form of the corruptions is unimportant. What we need to know is _which_ independent functions are used to generate the data rather than the exact way the functions are implemented. In terms of changing the semantic meaning of classes, in the limit of enough corruptions the class is guaranteed to change, as at some point the compositions will become noise and class identification will not be possible. This is why we focus most of our analysis in 4.3 on compositions of small number of corruptions. Whilst in the limit we cannot expect to always be compositionally robust, that does not make it meaningless to talk about robustness beyond single corruptions before we reach this limit.
>
>
> ---
>
> [1] Jacob Andreas, Marcus Rohrbach, Trevor Darrell, and Dan Klein. Neural module networks. In Proceedings of the IEEE conference on computer vision and pattern recognition, pp. 39–48, 2016.
>
> [2] Anirudh Goyal, Alex Lamb, Jordan Hoffmann, Shagun Sodhani, Sergey Levine, Yoshua Bengio, and Bernhard Schölkopf. Recurrent independent mechanisms. In International Conference on Learning Representations, 2021.
>
> [3] Akira Sakai, Taro Sunagawa, Spandan Madan, Kanata Suzuki, Takashi Katoh, Hiromichi Kobashi, Hanspeter Pfister, Pawan Sinha, Xavier Boix, and Tomotake Sasaki. Three approaches to facilitate invariant neurons and generalization to out-of-distribution orientations and illuminations. Neural Networks, 155:119–143, 2022.
>
> [4] Martin Arjovsky, Léon Bottou, Ishaan Gulrajani, and David Lopez-Paz. Invariant risk minimization. arXiv preprint arXiv:1907.02893, 2019.

---

> ### Author Response · Authors · 2024-04-09
> **Response now viewable**
>
> As with Reviewer XyV9, we added our response on 24 Feb but the default settings did not make it available to reviewers. Hopefully the response is now available to everyone

---

### Review · Reviewer_GWdu · 2024-03-18

**Summary Of Contributions:**

The paper under review investigates compositional robustness in image classification, and, in particular, the problem of maintaining high classification accuracy in the face of a composition of image corruptions obtained from the combination of elemental perturbations seen at training time. On this task, the paper  experimentally compares multiple architectures and learning paradigms: empirical risk minimization, an invariance promoting contrastive approach, and a novel modular architecture proposed to better capture an "inductive bias" that aligns with the compositional structure of the task.
Correspondingly, the modular architecture outperforms the other approaches in terms of displaying better generalization to compositions of image corruptions.
In addition, the paper finds that the degree of invariance between representations of 'in-distribution' elemental corruptions does not seem to correlate with robustness to 'out-of-distribution' compositions of corruptions, which might throw into question the efficacy of promoting invariance to particular transformations as an inductive bias, in particular in the hope achieving invariance to the combined transformations.

**Audience:**

Yes

**Broader Impact Concerns:**

The paper does not seem to raise any broader impact concerns.

**Claims And Evidence:**

No

**Requested Changes:**

* The paper would benefit from a more formal definition of the compositional nature of the task, and in particular, a more precise definition of what constitutes an elemental corruption, and how the proposed corruptions can be meaningfully thought of inducing a compositional structure, even seemingly without the properties that are typically associated with compositionality such as productivity and systematicity.
* The authors might want to consider pulling the appendix figure illustrating the modular architecture up into the main text, as it is a key part of the paper's contribution (assuming that space constraints might allow for that).
* The facts that in Figure 3 for the modular architecture on EMNIST elemental invariance is statistically significantly negatively correlated with accuracy and compositional invariance is a bit puzzling, quickly mentioned in the paper but not discussed.

**Strengths And Weaknesses:**

## Strengths
* The paper addresses an interesting problem in the context of robustness of neural networks to distributional shifts, and in particular, to compositions of elemental corruptions, which seems like a novel take.
* The paper explicitly draws a connection between compositionality, invariance and modularity, which, although intuitively connected, are not usually discussed and analyzed in this unified fashion.
* The paper introduces a novel compositional noise robustness benchmark which could constitute an interesting new setting to evaluate algorithms in the robustness and OOD literature.

## Weaknesses
* The concept of compositionally composing image corruption is interesting and intuitively intriguing, but it is unclear or at least formally ill-defined in what sense the proposed elemental corruption are indeed "elemental", i.e. in which sense they can be verified to be "atomic", "independent" or "orthogonal" among each other.
* This lack of formal definition of the elemental corruptions also translates in a difficulty in understanding the exact compositional nature of the proposed task, which for instance does not seem to naturally accommodate notions like systematicity and productivity which are commonly associated with compositional structures.
For instance, corruptions like *Contrast* and *Gaussian Blur* are such that multiple applications will progressively destroy information in the image (e.g. in these cases because multiple applications will result at some point in a uniform image). The fact that some chosen elemental corruptions will progressively irreversibly destroy information in the image seems to fly in the face of the notion that they afford compositional productivity, i.e. that one can combine an arbitrary sequence of corruptions resulting in a valid corruption, since the resulting at some point the combination will just result in the complete erasure of the image instead.
* Related and in addition to the general lack of formal definition of what constitutes an elemental corruption, it is noticeable that the image transformations that make up the proposed elemental corruptions have very distinct "group structure". For instance, *Rotate90* spans a discrete group, while other corruptions are less structured, which again seems to question the notion that they underly a bona fide compositional structure, i.e. that addressing the result of their combination can be done by combining the ways to address them individually.
In light of these doubts, one of the main results of the paper, i.e. that combining invariances to the individual corruptions is not enough to address the combination of corruptions, might not come as a surprise.
* The modular architecture has the arguable weakness that it requires knowledge of the specific composition of corruptions that is being applied on the image in order to undo it. This seems impractical in a concrete setting where one might just be presented with an image perturbed by an unknown corruption.

---

> ### Author Response · Authors · 2024-04-04
> **Updates & Clarifications (1 of 2)**
>
> Thank you for your detailed review, we are pleased you find our problem setting and unified approach interesting. Below we respond to the specific points for improvement and provide additional clarifications:
>
> ---
>
> "The concept of compositionally composing image corruption is interesting and intuitively intriguing, but it is unclear or at least formally ill-defined in what sense the proposed elemental corruption are indeed "elemental", i.e. in which sense they can be verified to be "atomic", "independent" or "orthogonal" among each other."
>
> The easiest formalisation would be a probabilistic one where our compositional dataset has joint distribution $P(X, C_1, ... C_6)$, with $C_i$ a binary random variable representing the presence of the corruption of not. The corruptions are "elemental"/"independent" in the sense that this distribution can be factorized as $P(X|C_1,...,C_6)\prod_i P(C_i)$ or alternatively, $C_i$ and $C_j$ are independent $\forall i \neq j$. In our setup this is true by construction. To generate the training set we start with the case $C_i=0 \ \forall \ i$, then "switch on" only individual $C_i$, that is, training also on the elemental cases where $C_i=1$ and $C_j=0, \ \forall \ i \neq j$. This is one sense in which the corruptions can be seen as "atomic"/"orthogonal".
>
> ---
>
> "Difficulty in understanding the exact compositional nature of the proposed task, which for instance does not seem to naturally accommodate notions like systematicity and productivity which are commonly associated with compositional structures."
>
> We view the task in a slightly different way. Rather than assuming that any image under any composition of corruptions is valid data we assume every identifiable instance of a class can be constructed from a composition of elemental factors. That is, there exists data that can be viewed as a composition of underlying "elemental" factors, but that is not to say that any arbitrary combination of these underlying factors creates valid or identifiable data. Whilst we may be able to recognize objects under specific combinations of corruptions that have never been seen before this does not mean we can recognize objects under every possible combination of corruptions.
>
> As you say, repeated application of noise, or too many destructive corruptions will make the data not possible to classify. This paper is concerned with the space between single corruptions and unrecognizable data, that is, the space where we can compose corruptions and the data remains identifiable. We simulate such a system, beginning by selecting the generative factors (elemental corruptions) used to generate the data.
>
> We can also provide some additional context from the literature:
>
> _From philosophy/linguistics:_
> Taking an example from the [systematicity of language](https://plato.stanford.edu/entries/compositionality/#Syst), we can understand "within an hour" and "without a watch" as valid sentences but can we also understand "within a watch" and "without an hour"? That is, even though language is largely compositional and systematic we can still compositionally construct gramatically correct phrases which do not have a well defined meaning.
>
> _From machine learning:_
> The opening of [Locatello et al's. work on disentanglement](https://arxiv.org/pdf/1811.12359.pdf) states "In representation learning it is often assumed that real-world observations x (e.g., images or videos) are generated by a two-step generative process. First, a multivariate latent random variable z is sampled from a distribution P (z). Intuitively, z corresponds to semantically meaningful factors of variation of the observations". Under this assumption the structure we are studying could be said to have semantically meaningful variables representing the class label and the corruptions that are present in an image.
>
> Alternatively, section 4 of [Scholkopf et al's. review on causal representation learning](https://arxiv.org/pdf/2102.11107.pdf) covers independent causal mechanisms which assumes "[The generative process] is composed of autonomous modules that do not inform or influence each other. In the probabilistic case, this means that the conditional distribution of each variable given its causes (i.e., its mechanism) does not inform or influence the other mechanisms." Under this assumption, the structure we are studying uses the elemental corruptions as the autonomous modules that do not inform or influence each other.

---

> > ### Author Response · Authors · 2024-04-04
> > **Updates & Clarifications (2 of 2)**
> >
> > "it is noticeable that the image transformations that make up the proposed elemental corruptions have very distinct "group structure""
> >
> > We are not sure we fully understood the point being made but, by design, we include both corruptions that can be seen as members of a finite group (R90 $\in Z_4$, invert $\in Z_2$) and corruptions that cannot (blur and noise are non-invertible). We choose to do this because there is a lot of existing work on group invariance and equivariance which, in our view, does not capture the reality that data can be corrupted by non-invertible/non-group-action functions. This is discussed further in the related work, subsection "Relational Inductive Biases and Modularity".
> >
> > ---
> > "The modular architecture has the arguable weakness that it requires knowledge of the specific composition of corruptions that is being applied on the image in order to undo it. This seems impractical in a concrete setting where one might just be presented with an image perturbed by an unknown corruption."
> >
> > We totally agree, our proposed method isn't practical. However, our aim is not to provide a practical method, but rather to ask, if the data has this structure, how should our architectures to model the data be designed and what sort of inductive biases should we apply?
> >
> >
> > ---
> > Requested Changes:
> > "a more formal definition of the compositional nature of the task"
> > - Hopefully our clarifications above have helped to provide a more formal description
> >
> > "pulling the appendix figure illustrating the modular architecture up into the main text"
> > - We will do this. Both you and Reviewer Jocc indicate it would improve readability (we have not done this yet to respond as quickly as possible to your other comments)
> >
> > "The facts that in Figure 3 for the modular architecture on EMNIST elemental invariance is statistically significantly negatively correlated with accuracy and compositional invariance is a bit puzzling"
> > - Yes it is a bit puzzling and we don't have a good explanation for which we have evidence. We don't see this negative correlation with modular architectures for other datasets. On the other hand for CIFAR10 we see a positive correlation with ERM. Also for CIFAR10 for some seeds the contrastive approach has positive correlation, for others it does not. Our only conclusion is that the transfer of invariance, and the correlation with accuracy, is neither consistent nor guaranteed across architectures and datasets

---

### Decision · Action_Editor_Bq9u · 2024-05-19

**Recommendation:** Reject

**Comment:**

The submission is not yet ready for TMLR in its current form primarily due to an inconsistency between the broad nature of the claims made and the evidence provided for these claims. A major revision to address this inconsistency would need to (1) more clearly scope the claims about the generality of the compositional robustness setting, and (2) provide more robust evidence that it is modularity (rather than domain information) that aids in this setting.

Separately, some of the reviewers note that the availability of the oracle decomposition of a task at test time is unrealistic. My editorial opinion is that whether a task is too idealized is subjective and often cannot be evaluated independent of the insights that can be drawn from the idealized setting (*i.e.*, it cannot be an *a priori* refutation). Since other aspects of the submission were more significant for the decision, it was not necessary for me to adjudicate this subjective point. Nevertheless, I strongly suggest that the authors more clearly delineate the *supervised modularity* (like neural module networks or conditioning) and *unsupervised modularity* (like emergent modularity or mixture of experts) approaches when introducing the present modular approach relative to prior work.

**Audience:**

The finding that supervised modularity aids in this compositional task may be of interest as a confirmation in another setting of prior related works that employ an oracle decomposition at test time to compose elemental modules (incl. the neural module network of Andreas et al., 2016b, and other works cited in the submission). However, restricting the scope of the claims to the image corruption task and to an *ad hoc* modular approach, as is appropriate to match the claims with the existing evidence, would significantly limit the interested audience.

**Claims And Evidence:**

Not entirely, as the evidence is insufficient for the broad scope of two principal claims:

1. **Generality of composed corruptions**: The robustness test to compositions of image corruptions is claimed to stand in for a more general setting "...with a compositional structure where the underlying elemental functions are known..." which would allow us "...to better understand how neural networks behave on out-of-distribution compositional data." The findings in Section 4 are taken as evidence that "...if such a set [of elemental transformations from which all visual stimuli can be composed] exists, the ideas presented in this work suggest that modular architectures may be able to model this space more efficiently than large monolithic models." The conclusion states that "...[the] work represents only a first step in understanding how neural networks behave under compositional structures."

    However, as reviewers note, the compositionally corrupted image data may not represent the broader class of compositional data considered in the prior work. This is for several reasons: they may not be naturalistic (in the sense of generating natural observables), they are a specific discrete set of transforms applicable to specific data, and some of them are not invertible (by construction, and also see Figure 2, where 5 or more image corruptions classification accuracy to chance even for the modular architecture). An adjustment to this claim to track the scope of the evidence—namely that the submission "investigates robustness to a sequence of image corruptions" and that the task is compositional only in the restricted sense of composing these corruptions—would be more appropriate.

2. **Modularity improves performance in compositional settings:** A significant claim is: "...for compositional robustness, when training domains consist only of the elemental components, modular approaches tend to outperform monolithic (non-modular) approaches." However, a single approach to modularity is studied, this modular approach appears to be tuned for this task alone, and the demonstration that this modular approach aids in this setting does not stand on strong footing (due to modest robustness and high variance in performance in Figure 2 and Table 1, as noted by the authors in Section 4.4) as compared to demonstrations of modular approaches in analogous supervised-compositional settings (e.g., Andreas et al., 2016b and its lineage).

    Additionally, the modular approach is provided with more information than the non-modular approaches, namely the sequence of corruptions applied to the test data. As such, there is a confound between modularity and extra supervision in the presented experiments. I disagree with the authors' statement during the discussion that "...there is a correspondence between modular architectures and knowledge of which corruptions (or mechanisms) are present" and that it is not clear "how to provide the corruption information without creating a modular architecture." As one reviewer hinted at during the closed recommendation period, there are many non-modular approaches that use conditioning (also called modulation) to incorporate domain (or temporal or positional) information that could have been employed here as a baseline to disentangle modularity and extra supervision (e.g., [Perez et al., 2018](https://arxiv.org/abs/1709.07871); [Monteiro et al., 2021](https://arxiv.org/abs/2106.13899)). Such a baseline would be necessary to establish the strong claim that modularity helps in this setting.

    In contrast, an adjustment in claim to "extra domain information plus this modular architecture helps to a modest degree at the cost of increased variance" would also close the gap between the claims and evidence; however, the consequence would be a diminished audience for this adjusted claim (see "Audience" below).

**Resubmission Of Major Revision:**

The authors may consider submitting a major revision at a later time.